# Isoprene emission and photosynthesis during heat waves and drought in black locust

Ines Bamberger[1], Nadine K. Ruehr[1], Michael Schmitt[1], Andreas Gast[1], Georg Wohlfahrt[2], Almut Arneth[1]

[1]Institute of Meteorology and Climate Research - Atmospheric Environmental Research, Karlsruhe Institute of Technology (KIT/IMK-IFU), Garmisch-Partenkirchen, Germany
[2]Institute of Ecology, University of Innsbruck, Innsbruck, Austria

*Correspondence to:* Ines Bamberger (ines.bamberger@kit.edu)

**Abstract.** Extreme weather conditions, like heat waves and drought, can substantially affect tree physiology and the emissions of isoprene. To date, however, there is only limited understanding of isoprene emission patterns during prolonged heat stress and next-to-no data on emission patterns during coupled heat–drought stress, or during post-stress recovery. We studied gas exchange and isoprene emissions of black locust trees under episodic heat stress and in combination with drought. Heat waves were simulated in a controlled greenhouse facility by exposing trees to outside temperatures +10°C, and trees in the drought treatment were supplied with half of the irrigation water given to heat and control trees. Leaf gas exchange of isoprene, $CO_2$ and $H_2O$ was quantified using self-constructed, automatically operating chambers, which were permanently installed on leaves (n=3 per treatment). Heat and combined heat–drought stress resulted in a sharp decline of net photosynthesis ($A_{net}$) and stomatal conductance. Simultaneously, isoprene emissions increased six- to eight-fold in the heat and heat–drought treatment, which resulted in a carbon loss that was equivalent to 12 % and 20 % of assimilated carbon at the time of measurement. Once temperature stress was released at the end of two 15-days-long heat waves, stomatal conductance remained reduced, while isoprene emissions and $A_{net}$ recovered quickly to values of the control trees. Further, we found isoprene emissions to co-vary with $A_{net}$ during non-stress conditions, while during the heat waves, isoprene emissions were not related to $A_{net}$ but to light and temperature. Under standard air temperature and light conditions (here 30°C and photosynthetically active radiation of 500 µmol m$^{-2}$s$^{-1}$), isoprene emissions of the heat trees were by 45 % and the heat–drought trees were by 27 % lower than in control trees. Moreover, temperature response curves showed that not only the isoprene emission factor changed during both heat and heat-drought stress, but also the shape of the response. Because introducing a simple treatment-specific correction factor could not reproduce stress-induced isoprene emissions, different parameterizations of light and temperature functions are needed to describe tree isoprene emissions under heat and combined heat–drought stress. In order to increase the accuracy of predictions of isoprene emissions in response to climate extremes, such individual stress parametrizations should be introduced to current BVOC models.

## 1 Introduction

Under a warming climate, extreme weather conditions, like heat waves and drought, are observed to occur more frequently (Coumou and Rahmstorf, 2012). Forested ecosystems contribute the majority of the global emissions of volatile organic compounds to the atmosphere (Guenther et al., 2012) and these emissions are expected to change with increasing frequency and intensity of climate extremes (Staudt and Peñuelas, 2010), which might

persist following stress release. Up to now, however, there is only limited understanding of biogenic volatile organic compound (BVOC) emissions from trees during prolonged heat and combined heat–drought stress, including emission patterns during post-stress recovery.

With annual estimates ranging from 350 Tg yr$^{-1}$ to 800 Tg yr$^{-1}$, isoprene contributes most to the global budget BVOC emissions (Guenther et al., 2012). Influencing tropospheric ozone and methane levels (Atkinson, 2000)
and the formation of secondary organic aerosols (Carlton et al., 2009; Wyche et al., 2014), isoprene plays an important role in atmospheric chemistry and has an indirect effect on climate. Global isoprene emissions are most often estimated using the so-called Guenther algorithms taking into account the temperature-, and light-dependence of emissions (Guenther et al., 1991, 1993). In these algorithms, a species-specific standard emission factor (E$_s$, a constant that describes leaf emissions at standard conditions of typically 30°C and a
photosynthetically active radiation of 1000 µmol m$^{-2}$ s$^{-1}$) is multiplied by temperature and light functions. Guenther algorithms have been successfully used to model isoprene fluxes at spatial scales ranging from ecosystem to the globe (e.g. Naik et al., 2004; Guenther et al., 2006, 2012; Lathière et al., 2006; Potosnak et al., 2013; Brilli et al., 2016). However, the temperature- and light-functions depend on empirically derived parameters, which may not be constant across different regions, or climatic conditions (Arneth et al., 2008;
Niinemets et al., 2010b). Moreover, E$_s$ is known to vary, even within a given species, for example in response to weather extremes (Niinemets et al., 2010a; Geron et al., 2016). Thus the modeling algorithms often fail to reproduce isoprenoid emissions of ecosystems under stress, irrespective of whether stress is induced mechanically or by drought (Kaser et al., 2013; Potosnak et al., 2013). Owing to the sparse amount of data, accounting for stress-induced BVOC emissions is one weak point of global BVOC models (Niinemets et al.,
2010a; Guenther, 2013) and calls for further research in this area.

Isoprene emissions by plants are constitutive and their emission pathway is relatively well known (Loreto and Schnitzler, 2010). Plants usually synthesize isoprene via the methylerithriol phosphate pathway (MEP) using carbon pools from photosynthesis (Sharkey and Yeh, 2001). Isoprene emission requires *de novo* synthesis, meaning that isoprene emissions from plants are predominantly dependent on enzymatic activity (Sharkey and
Yeh, 2001). As a consequence, isoprene emissions are usually both temperature, and light dependent (Niinemets et al., 2004). While under normal conditions only 1-2 % of the carbon fixed during photosynthesis is emitted as isoprene (Harrison et al., 2013) this fraction may increase to more than 50 % under stress (Sharkey and Loreto, 1993; Pegoraro et al., 2004b; Loreto and Schnitzler, 2010). Why do plants invest so much carbon to maintain isoprene production under adverse conditions? The importance of isoprene for plant functioning is not
completely resolved (Harrison et al., 2013), but several lines of evidence suggest that isoprene helps to protect the photosynthetic apparatus during oxidative and thermal stress (Velikova and Loreto, 2005; Behnke et al., 2007; Vickers et al., 2009; Ryan et al., 2014). Hence isoprene formation and emission may represent an important mechanism for heat and drought tolerance in deciduous trees, which needs further investigation.

Episodic environmental stress conditions caused by heat waves or soil water deficit are projected to increase
significantly in frequency and/or severity under a future climate (Coumou and Rahmstorf, 2012). Drought periods often coincide with high temperatures (Boeck and Verbeeck, 2011). While some efforts have been made to quantify isoprene emissions in presence of single stress factors like high temperature (Sharkey and Loreto, 1993; Singsaas and Sharkey, 2000) or soil water deficit (Pegoraro et al., 2004b; Brilli et al., 2007, 2013; Ryan et

al., 2014), there is up to now only one study that has studied the effects of combined prolonged heat and drought on trees (Vanzo et al., 2015). Additionally, most stress-response studies have been based on fixed environmental conditions, for instance applying discrete temperature steps, and stress exposure was often limited to short time periods and only applied to specific plant tissues such as leaves. Although this makes it easier to disentangle the effects of each single stress factor, the plant emission response might differ compared to closer to natural conditions (e.g. fluctuating environmental conditions, prolonged stress exposure, Niinemets et al., 2010a, 2010b). A complete and quantitative understanding of the effects of prolonged temperature and/or soil moisture stress on isoprene emissions under naturally fluctuating conditions has not yet been reached.

Broadleaf deciduous tree species cover about one-third of the global land area, but are estimated to be responsible for the majority of global BVOC emissions (Guenther, 2013). Black locust (*Robinia pseudoacacia* L.), a deciduous tree and relatively strong isoprene emitter (Kesselmeier and Staudt, 1999), originally native to North America, is nowadays quite commonly planted in Europe (Cierjacks et al., 2013). Due to its rapid growth and its comparatively high tolerance to stress (e.g. drought stress, Mantovani et al., 2014; Ruehr et al., 2016) the area where the tree species is grown is expected to further increase under a warmer climate (Kleinbauer et al., 2010).

Here we aim to evaluate the effects of prolonged and combined heat and drought on isoprene emissions of black locust trees. The study was designed to alerting the modeling community to the complexity of response patterns when isoprene emissions are studied under close-to-natural conditions by mimicking outside temperature variability. The objectives of this study were, (1) to quantify heat and heat-drought impacts on isoprene emission rates of black locust trees and the isoprene emission response following recovery, (2) to gain more insight into the apparent fraction of photosynthetic carbon equivalents used for isoprene emission during stress, and (3) to evaluate empirical temperature and light response curves of isoprene emission rates under prolonged exposure to heat and heat-drought stress. A greenhouse experiment with two week-long heat waves (+10°C above outside ambient temperatures) was conducted, followed by a recovery period of one week mimicking outside temperatures. During the experiment, isoprene emission rate of black locust trees was measured concurrently with the $CO_2$ and $H_2O$ gas exchange using an automated leaf chamber setup.

## 2 Materials and methods

### 2.1 Experimental set-up

Black locust seedlings (*Robinia pseudoacacia* L.) were grown in a controlled greenhouse facility in Garmisch-Partenkirchen, Germany (708 m a.s.l.) The trees had been planted in individual large pots (120 l) filled with a mixture of humus and sand (ratio of 2:3) in September 2012. In 2014, when the experiment was conducted, trees were four-years old. As the experiment was part of a three-years campaign, which sought to evaluate the long-term effects of repeated heat waves and heat-drought waves on tree growth and performance, the trees in the

stress treatments had already been exposed to two experimental heat/heat-drought waves during summer 2013. The basal area in the previously stressed trees was slightly lower than that of the control trees before the experiment was initiated in 2014 (heat: -13% and heat-drought -16%), although, basal growth rates and photosynthesis recovered to pre-stress conditions three weeks after stress relief in 2013 (Ruehr et al., 2016). After the experiment in 2013 ended, the trees were pruned to 1.80 m height, kept outside during winter, and

transferred back into the greenhouse in May 2014, where the measurements were performed from 10th of July until 26th of August.

Black locust trees were kept in two adjacent neighboring, separately controllable compartments of the greenhouse facility from May onwards. The environmental conditions in the greenhouse (equipped with UV-transmissive glass) were regulated by a computer (CC600, RAM Regel- und Messtechnische Apparate

GmbH, Herrsching, Germany) and air temperature settings followed outside conditions measured in front of the greenhouse, while relative humidity was set to mimic the diurnal of long-term (20-year) monthly averages from a meteorological station close-by. Photosynthetically active radiation (PQS1, Kipp & Zonen, Delft, The Netherlands), air temperature and relative humidity (CS251, Campbell Scientific Inc., Logan, UT, USA) in each greenhouse compartment were monitored by two sensors each. During non-stress conditions, differences in

environmental conditions between the two compartments of the greenhouse were generally small during both years of the experiment (Ruehr et al. 2016, Duarte et al., 2016). Prior to the first heat wave, the black locust trees grew under the same environmental conditions from 7th of May until 13th of June and none of the environmental drivers differed by more than 2% (see Table 1). While in one compartment of the greenhouse, trees were always kept under the ambient conditions as described above (control trees, n=6), trees in the second compartment (heat

and heat–drought trees, n=12) were periodically exposed to two consecutive heat waves simulated by a +10°C increase of temperature lasting between 14 to 15 days. During the heat waves, relative humidity in the heat compartment of the greenhouse was controlled to decrease so that vapor pressure deficit increased. Each heat wave was followed by a recovery phase of 7 days. While control and heat-treated trees received on average 2.6 l tree$^{-1}$ day$^{-1}$ irrigation, heat–drought treated trees received (starting 6 days before heat stress) only 1.3 l tree$^{-1}$

day$^{-1}$. Recovery periods were initiated by supplying each tree once with a larger amount of water (10.8 l) which increased soil moisture and largely reduced soil water deficit. Isoprene emissions were measured in parallel with the $CO_2$ and $H_2O$ gas exchange using leaf chambers attached to three different trees per treatment. After the experiment, leaf biomass within each chamber was harvested, dried and weight and half sided leaf area determined via previously determined specific leaf area. In one case, when an enclosed leaf wilted and dried, the

corresponding chamber was installed on an intact leaf of the same tree. To determine leaf biomass losses, leaf litter was collected, dried and weight. Leaf area was calculated from dry weight and treatment-specific leaf area

(data not shown).

The bottom of each tree pot was equipped with a coiled water pipe to provide soil cooling to mimic pre-defined soil temperatures at a depth of 50 cm (corresponding to air temperature averaged over the previous 20 days). Soil water content (10HS, Decagon Devices, Inc, Pullman, WA, USA) and soil temperature (T107, Campbell Scientific Inc., UT, USA) were measured in each pot at a depth of 10 cm and in additional pots at 30 and 50 cm depth. The volumetric soil water content ($SWC$) was used to determine the daily relative extractable soil water ($RSW$, in %) according to following relationship

$$RSW = 100 \times \frac{SWC - SWC_{min}}{SWC_{max} - SWC_{min}}, \tag{1}$$

with, $SWC_{min}$ and $SWC_{max}$ being experimentally derived minimum values of daily soil water content at 30 cm depth during drought and maximum values of mean daily $SWC$ per sensor. In order to get an average value per tree pot, $RSW$ from three different depths were averaged.

### 2.1.1 Automated leaf chamber set-up

The gas exchange of black locust leaves was measured using a self-made, automated chamber system on three trees per treatment (n=3), and one empty chamber as background reference. The chambers were constructed from a transparent cylinder, enclosed by two caps (inner volume: 6.65 l) all made of acrylic glass (PMMA, Sahlberg, Feldkirchen, Germany) coated with a FEP (fluorinated ethylene propylene, PTFE Spezialvertrieb, Stuhr, Germany) foil to ensure chemical inertness of the interior of the chamber. Gas leakage was minimized by sealing with PTFE foam, transparent tape and plastic sealing band (Teroson, Düsseldorf, Germany) between branch, lids and chamber body. During the experiment, a 12 V fan (412 FM, EBM-Papst, Mulfingen, Germany) was constantly running in each chamber to provide homogeneous mixing inside the plant chamber. The chambers were permanently installed and enclosed one black locust leaf (average leaf area of 129 cm$^2$), which was inserted via the tree-facing side of the chamber, which could be easily taken apart and the leaf petiole inserted. A second cap, facing-away from the tree, was typically held open and closed only during the measurement time for 8 to 10 min (using pressurized air) when the chamber was supplied with an external air stream. Opening and closing of the chambers was automatically controlled.

To produce $CO_2$, water vapor and VOC and $O_3$ free zero air for the external air stream, outside air was drawn by an oil free scroll compressor (SLP-07E-S73, Anest Iwata, Japan) through an Ultra Zero Air Generator (N-GT 30000, LNI Schmidlin SA, Geneve, Suisse). In parallel, a second air stream (Liquid Calibration Unit, Ionicon, Innsbruck, Austria) added $CO_2$ and $H_2O$ to the zero air at a rate of 1 nL min$^{-1}$ (normalized liter per minute). Together a constant flow of 7 sl min$^{-1}$, containing $409 \pm 11$ µmol mol$^{-1}$ $CO_2$, $6.1 \pm 0.4$ mmol mol$^{-1}$ $H_2O$, and VOC free air, was routed through a chamber during the measurement (Fig. 1).

The main tubing line of the chamber set-up was 3/8 inch stainless steel tubing (Swagelok, Ohio, USA) coated with SilcoNert (Silco Tek GmbH, Bad Homburg, Germany). The direction of air flow to the different chambers was controlled by 2/2 way solenoid valves with PTFE housing (0121-A-6,0-FFKM-TE, Bürkert, Ingelfingen, Germany) connected by a 3/8 inch PTFE tube (ScanTube GmbH, Limburg, Germany) to the inlet and outlet of

the leaf chambers (Fig. 1). Another valve placed in the center of the main tubing (see Fig. 1 Vmain) could be opened to flush the entire system with VOC-free air.

During the automatic switching between the individual leaf chambers and the empty chamber (performed by controlling valves and fans via ICP modules, I-7067D, ICP DAS, Hsinchu County, Taiwan), each chamber was sampled for at least eight minutes. Between the measurements of different chambers the tubing was flushed with the VOC-free synthetic air for one minute.

        Due to the extended leaf area and because black locust leaves are known to fold their leaves during night time
and during excessive temperature stress (paraheliotropism) it was not possible to install a leaf temperature sensor. Leaf chamber air temperature was measured by a thermocouple (5SC-TT-TI-36-2M, Newport Electronics GmbH, Deckenpfronn, Germany) in each chamber, and light conditions were recorded by a photodiode optimized to measure photosynthetically active radiation (G1118, Hamamatsu Photonics, Hamamatsu, Japan). Additionally, a comparison between air temperatures and leaf temperatures during the
second heat wave showed that leaf temperature was (independent of the treatment) not significantly different from air temperature (Appendix A and Table A1). Photodiodes were cross-calibrated using a photosynthetically active radiation (PAR) sensor (PQS 1, Kipp & Zonen, Delft, The Netherlands).

**2.1.2 Water vapor and carbon dioxide exchange**

        Concentrations of $CO_2$ and $H_2O$ in the ingoing and outgoing airstream were measured by a Li-840 (for absolute concentrations) connected to a Li-7000 (LI-COR Inc., Lincoln, NE, USA) running in differential mode (Fig. 1). This allowed measuring differences between ingoing and outgoing air concentrations, as well as absolute concentrations. The three measurement cells of both infrared gas analyzers (IRGA) were supplied with 0.5 l min$^{-1}$
each, provided by a pump (NMP830KNDC, KNF, Freiburg, Germany), connected to a mass flow controller (F-201CV-1K0-RAD-22-V, Bronkhorst, Ruurlo, NL). The two measurement cells of the Li-7000 were matched regularly and recalibrated with the Li-840 on a bi-weekly basis. To detect and remove any offsets not influenced by plant gas exchange between outgoing and ingoing air, measurements of an empty chamber were performed. To calculate gas exchange rates, we determined average concentration differences (air$_{delta}$= air$_{out}$- air$_{in}$) under
steady state conditions (between 300s and 490 s after chamber closure). Steady state criteria were reached when the standard deviation of averaged differences (within the above defined time frame for steady state) in water vapor ($\Delta H_2O$) was < 0.5 mmol mol$^{-1}$ and the rate of change in $\Delta H_2O$ over time was < 0.01 mmol mol$^{-1}$s$^{-1}$. Transpiration ($Tr$ in mmol m$^{-2}$ s$^{-1}$) was calculated using the following equation

$$Tr = \frac{f\Delta H_2O}{l_a\left(1-\frac{H_2O_{out}}{1000}\right)},$$
(2)

where $f$ is the air flow rate in mol s$^{-1}$, $\Delta H_2O$ the difference in water vapor between ingoing and outgoing air ($H_2O_{out}$) in mmol mol$^{-1}$, and $l_a$ the half-sided leaf area in m$^2$.

        Net photosynthesis ($A$ in µmol m$^{-2}$ s$^{-1}$) was assumed to reach steady state when the standard deviation of the averaged differences (within the above defined time frame for steady state) in $CO_2$ between ingoing and

outgoing air ($\Delta CO_2$) was <2.5 μmol mol$^{-1}$ and the rate of change in $\Delta CO_2$ was < 0.2 μmol mol$^{-1}$s$^{-1}$, and was then derived as follows

$$A_{\text{net}} = \frac{f\Delta CO_2}{l_a} - \frac{(CO_{2\,\text{out}}Tr)}{1000},$$ (3)

where $CO_{2\,\text{out}}$ is the $CO_2$ concentration of the outgoing air in μmol mol$^{-1}$ corrected for dilution by transpiration. Additionally, the $CO_2$ background of the empty chamber was removed in order to correct for chamber effects on $CO_2$ mixing ratios. Equation (3) results in net photosynthesis ($A_{\text{net}}$) calculated by accounting for the dilution by transpiration.

Stomatal conductance ($g_S$ in mol m$^{-2}$s$^{-1}$) was calculated from transpiration using the following formula

$$g_S = \frac{Tr\left(1000 - \frac{W_L + H_2O_{\text{out}}}{2}\right)}{W_L - H_2O_{\text{out}}},$$ (4)

with, $W_L$ referring to the mole fraction of water vapor in the leaf (in mmol mol$^{-1}$) as calculated from the ratio of the saturation vapor pressure at a given leaf temperature and the atmospheric pressure (both given in kPa).

Mid-day leaf water potential was measured by determining the pressure necessary to cause water to exclude from a freshly-cut leaf inserted in a Scholander pressure chamber (Model 1000, PMS Instrument Company, Albany, Oregon, USA).

### 2.1.3 Volatile organic compounds

Measurements of isoprene emissions were performed using a high sensitivity proton-transfer-reaction mass spectrometer (PTR-MS, IONICON, Innsbruck, Austria) operated at a drift tube pressure of 2.3 mbar, and a temperature and voltage of 60°C and 600 V along the drift tube, respectively. The operation principle of the PTR-MS is described elsewhere (Hansel et al., 1995; Lindinger et al., 1998). The PTR-MS was operated to sequentially measure a set of preselected mass channels (assignable to BVOCs) including isoprene (m/z 69).

At regular intervals, calibrations of the PTR-MS at ambient humidity were conducted routing an air mixture containing several volatile organic compounds at predefined mole fractions of 7 ppb, 10 ppb, 15 ppb and 20 ppb through the instrument (Fig. 1). The air mixture was provided by a liquid calibration unit diluting a gas standard (IONICON, Innsbruck, Austria) containing 15 different volatile organic compounds in $N_2$ at ppm levels with VOC-free zero air. During the measurements in 2014, the sensitivity for isoprene was determined to be between 7.0 and 7.3 ncps ppb$^{-1}$ (normalized counts per second and ppb, normalized to a drift tube pressure of 2.2 mbar and 1 million primary ions). The limit of detection for isoprene was determined to be around 0.4 ppb at an integration time of one second. Although a potential interference of the isoprene signal on nominal mass to charge ratio m/z =69$^+$ with C5 green leaf volatiles (e.g. methylbutanal, methylbutenol, or pentenol) appearing at the same mass to charge ratio is possible, we argue for such an interference to be unlikely because until now, such compounds were only observed following artificial cutting and drying of leaves from plant species which are not emitting isoprene (Fall et al., 2001). In addition, within the suit of measured volatiles we found no indication for other compounds, which should co-occur with such C5-volatiles (e.g. C6 leaf alcohols or

acetaldehyde). In a similar study the absence of those compounds during a heat-drought wave was referred to that fast drying after artificial cutting is not comparable to natural drought progression (Vanzo et al., 2015).

The isoprene flux ($E_{iso}$ in nmol m$^{-2}$s$^{-1}$) was calculated according to Niinemets et al. (2011)

$$E_{iso} = (c_{out,c} - c_O) \frac{f}{l_a}. \tag{5}$$

Where $c_{out,c}$ is the VOC concentration (in nmol mol$^{-1}$) measured at the outlet of the branch chamber, $c_0$ is the VOC concentration measured at the output of an empty chamber. With the subtraction of the VOC concentration measured at the outlet of an empty chamber ($c_0$), fluxes were corrected for the VOC background in the zero air

and possible fluxes from/to the empty chamber and the associated tubing. The empty chamber background for isoprene contributed on average only two percent to the total isoprene signal measured in the control plant chambers. Since the transpiration correction (rightmost term of Eq. (3)) for the control, heat and heat–drought chambers contributed on average less than 0.5 % of the daytime (PAR > 50 µmol m$^{-2}$s$^{-1}$) isoprene emissions, it was neglected. Isoprene concentrations reached their equilibrium usually about one minute later than $CO_2$ and

$H_2O$ concentrations and VOC measurements showed a larger level of noise compared to $CO_2$ and $H_2O$ concentrations, so the quality criteria for isoprene differed from the criteria for $CO_2$ and $H_2O$ exchange. To avoid systematic errors due to an insufficient air exchange in the chambers only isoprene concentrations during the last minutes of each chamber closure (after 360 s of closure until the end) were averaged to calculate the equilibrium isoprene fluxes (Eq. (5)). Measurements (a) when chambers were not closed sufficiently long (less than 420 s),

(b) when the performance of the PTR-MS was inadequate (e.g. directly after refilling the water bottle), or (c) when no empty chamber measurements were available, were discarded.

## 2.2. Modeling the temperature and light responses of isoprene

Since isoprene emissions from plants are temperature and light dependent, leaf-level isoprene fluxes can be

estimated from a light-dependent function $f_Q$, a temperature dependent function $f_T$, and an isoprene emission factor $E_S$, which is assumed to be a constant, but plant-specific factor which describes the isoprene emissions at reference conditions (e.g., a temperature of 30°C and PAR=500 µmol m$^{-2}$s$^{-1}$).

$$E_{iso} = E_S \, f_Q \, f_T \tag{6}$$

Temperature and light response functions are usually normalized to unity at standardized conditions and describe the shape of the isoprene emission curve. The response functions were first developed by Guenther et al. (1991, 1993). These models use a hyperbolic function to describe the light response function as follows

$$f_Q = \frac{C_{L1} \alpha Q}{\sqrt{1 + \alpha^2 Q^2}}, \tag{7}$$

where $C_{L1}$ is a scaling constant and $\alpha$ the quantum yield of isoprene emission; here, both parameters were

optimized for each treatment separately in order to best describe the light response function of the measured isoprene fluxes.

The temperature dependence of isoprene emissions is usually characterized by an exponential increase with leaf temperature until an optimum temperature $T_{opt}$ is reached followed by a subsequent exponential decrease (Guenther et al., 1991, 1993).

$$f_T = \frac{e^{\left(\frac{C_{T1}(T_L - T_S)}{R T_S T_L}\right)}}{1 + e^{\frac{C_{T2}(T_L - T_{opt})}{R T_S T_L}}},$$ (8)

$T_s$ is a standard temperature (usually 30°C, or 303 K) at which the normalized response curve is one, $R$ is the gas constant (8.314 J mol$^{-1}$ K$^{-1}$), $T_L$ is the leaf temperature in Kelvin and $C_{T1}$ and $C_{T2}$ are parameters which can be interpreted as activation and deactivation energy of isoprene emissions (in J mol$^{-1}$), respectively. Later on, a third parameter, $C_{T3,}$ was introduced to force the temperature response curve through one at the chosen standard temperature (Guenther et al., 2006). We did not normalize temperature and light response curves to one under standardized conditions and thus provide the original parameterized emission factor ($E_s$). However, to allow better comparisons with other studies, we tested if the original parameterized emission factor differed compared to the emission factor when the curve was forced through one and found that both values were within the given standard errors identical (maximal difference of 0.2 nmol m$^{-2}$s$^{-1}$). The experimentally derived average temperature response of the control, heat and heat–drought treated trees was used to optimize the parameters $C_{T1}$, $C_{T2}$ and $T_L$ using a nonlinear weighted fitting algorithm (see Section 2.4) for each treatment separately. For the control treatment we did not have enough data points in the high temperature range to constrain the optimum temperature, and therefore set the optimum temperature to a fixed value of 311.8 K (see Guenther et al., 1991) in order to optimize the remaining parameters.

## 2.3 Data analysis and statistics

The data post-processing and statistical calculations were performed using the commercial software package Matlab® (Version R2013b, Math Works®, MA, USA). To estimate leaf isoprene emissions we largely followed the standardization criteria (except for light control) for leaf-scale emission measurements recommended by Niinemets et al. (2011). Because temperature control was performed within the separate compartments of the greenhouse temperatures within the leaf chambers were recorded, but not controlled separately.

Increases or decreases in isoprene emissions and photosynthesis during the heat waves were calculated as treatment effect ($\frac{treatment - control}{control}$). To test for differences between treatments and time periods, we used a linear mixed effects model (using fixed effects for time period and treatment and random effects for tree and measurement day) to test for significant changes in daily average isoprene emissions for each treatment and during the different time periods of the experiment (Ruehr et al., 2016). A $p$-value $< 0.05$ was considered as statistically significant.

To determine light and temperature relationships of isoprene emissions in each treatment and during both stress periods, we grouped the data into 8 bins according to PAR levels (<5 µmol m$^{-2}$s$^{-1}$, 5–50 µmol m$^{-2}$s$^{-1}$, 50–100 µmol m$^{-2}$s$^{-1}$, 100–150 µmol m$^{-2}$s$^{-1}$, 150–200 µmol m$^{-2}$s$^{-1}$, 200–400 µmol m$^{-2}$s$^{-1}$, 400–650 µmol m$^{-2}$s$^{-1}$ and > 650 µmol m$^{-2}$s$^{-1}$) and temperature conditions (15-20°C, 20–24°C, 24–28°C, 28–32°C, 32–35°C, 35–40°C,

40–45°C and >45°) within the chambers. The parameterization of the light and temperature response functions (Eq. 7 and 8) was done for each treatment as follows. In an initial step, the bin-averaged isoprene data (PAR > 100 µmol $m^{-2}s^{-1}$ with temperature bins defined above) were fitted to the temperature-response function (Eq. 8) using a non-linear fitting algorithm weighted with the inverse standard deviation. The temperature fit (parameter $E_S$, $C_{T1}$, $C_{T2}$ and $T_m$) was then used to normalize the measured isoprene emissions to a standard temperature of 30°C before fitting isoprene data bin-avaraged for PAR to the light-response function (Eq. 7). The light-response fit (parameter $E_S \times C_{L1}$ and $\alpha$) was then used to normalize the measured isoprene flux data to standard light levels (PAR = 500 µmol $m^{-2}s^{-1}$), which were then again fitted to the temperature-response function. This procedure was repeated in an iterative way until all fitting parameters changed by less than 1 % between subsequent iterations. Since we deliberately included ambient diurnal and day-to-day temperature variations in our study, it was not possible to keep the number of data points constant in each temperature bin and treatment. To warrant comparability among the bin averages of each treatment, the same temperature bins were used in each treatment. As a consequence some bins at the upper end of the temperature distribution included only few data points, some only one. In the case the standard deviation could not be calculated, we assumed a conservative value of 100 (7-times higher than the highest standard deviation observed) to reduce the weight of this point for the temperature fit.

To exclude nighttime fluxes, which were always zero in all treatments, daytime averages were calculated exclusively for data when PAR was higher than 50 µmol $m^{-2}s^{-1}$. The fraction of recently assimilated carbon emitted as isoprene was calculated by dividing the isoprene carbon flux by the assimilated carbon and calculating bin averages after classifying the isoprene fluxes into 8 temperature bins as mentioned above.

## 3 Results

### 3.1 Environmental conditions

With a maximum of 34.7°C and a minimum of 27.8°C, daytime average air temperatures during the heat waves were considerably warmer than under ambient conditions (daytime-averaged maximum 23.2°C and minimum 13.7°C). Along with warmer temperatures, daily-averaged vapor pressure deficit (VPD) increased up to 3.0 kPa, while it remained below 1.0 kPa under ambient conditions (Fig. 2b). Outside the stress periods, air temperature and VPD did not differ between the greenhouse compartments (Fig. 2a,b).

While the daily-averaged relative extractable soil water content (RSW) remained above 40 % in the control trees, it decreased to 20 % in the heat and 15 % in the heat–drought treatment (Fig. 2c). After watering at the first day of the recovery, the soil water content in the stress treatments increased considerably and RSW during the recovery remained between 50 % and 70 % in both stress treatments. During the heat periods, RSW of the heat-stressed trees was slightly higher than RSW of heat–drought treated trees. Heat and heat–drought stress caused a large decline in midday leaf water potential (-1.7 MPa in the control compared to -2.3 MPa in both stress treatments, data not shown). In both treatments, we observed pronounced leaf shedding during the first heat wave and estimated that about 80 % of the leaves were shed in the heat treatment and 90 % in the heat–drought treatment, averaging in both treatments to a leaf area of about 2.4 $m^2m^{-2}$ lost. Leaf shedding reduces water loss and protects the integrity of the hydraulic system in black locust (Ruehr et al. 2016). The relatively larger leaf shedding in the heat-drought trees together with a smaller biomass (indicated by reduced basal area due to last year's experiment, see Methods) than under heat-stress only, might explain the small differences in RSW between the heat and heat-drought treatment (Fig. 2c), although irrigation in heat-drought trees was halved compared to the heat and control trees.

### 3.2 Stomatal conductance, net photosynthesis and isoprene emissions

Along with increased temperatures and reduced RSW during the stress periods, average daytime stomatal conductance in heat–drought stressed trees decreased to values below 0.01 mol $m^{-2}s^{-1}$ (Fig. 3a). The stomatal conductance of heat trees during the heat waves was higher compared to heat–drought trees, with daily averages between 0.01 and 0.03 mol $m^{-2}s^{-1}$, but still lower than in the control trees (Fig. 3a). Compared to the control, net photosynthesis during the first heat wave decreased on average by 44 % in the heat and 67 % in the heat–drought treatment (Fig. 3b). During the second heat wave, this decrease was smaller with 41 % in the heat and 46 % in the heat-drought treatment. Following stress release net photosynthesis in heat and heat-drought trees recovered quickly and reached values similar to the control trees after a few days. A linear mixed effects model comparing net photosynthesis during the stress periods to pre-stress control conditions confirmed the significance of these changes (Table 2), and that before the first heat wave, net photosynthesis did not differ significantly among treatments (see pre-treatment values, Table 2).

Daytime isoprene emissions of black locust in the heat treatment were on average by 153 % and 142 % higher than in the control trees during the first and second heat wave, respectively (Fig. 3c). In the heat-drought stressed trees isoprene emissions were 171 % higher than in the control trees during the first heat wave, and 333 % during

the second heat wave. During both recovery periods, isoprene fluxes decreased rapidly to values comparable to pre-stress conditions, suggesting a quick and complete recovery. Except for the heat-drought trees during the second heat wave the significance of changes in isoprene emissions in the heat and heat–drought treatment during stress phases was confirmed by a linear mixed effects model (Table 2), while before stress and during the two recovery phases isoprene emissions did not differ significantly among treatments and control (Table 2).

**3.3 Relationship between $CO_2$ and isoprene emissions under stress conditions**

We found isoprene emissions of the heat and heat–drought treated trees during the recovery periods to be clearly related to net photosynthesis ($A$) following an exponential function $E_{iso}=\exp(a*A)$-b (p-value<0.05; Fig. 4). Such a relationship was also visible in control trees as long as temperatures did not exceed 30°C (Fig. 4). In control trees, net photosynthesis was on average 4.5 µmol m$^{-2}$s$^{-1}$ and isoprene emission 1.5 nmol m$^{-2}$s$^{-1}$. During the heat waves, isoprene emission was not related to net photosynthesis in heat and heat–drought treated trees. Net photosynthesis decreased to 2.5 µmol m$^{-2}$s$^{-1}$ on average in the heat and 2.1 µmol m$^{-2}$s$^{-1}$ on average in the heat–drought treatment, while isoprene fluxes increased sharply to 11.2 nmol m$^{-2}$s$^{-1}$ in the heat and 5.9 nmol m$^{-2}$s$^{-1}$ in the heat–drought treatment on average.

In the temperature range from 28°C to 32°C, control trees emitted an equivalent of 1.6 % of assimilated carbon as isoprene (Table 3). In the same temperature range, heat and heat–drought stressed trees emitted equivalent to 0.8 % (t-test compared to control resulted in p < 0.05) and 1.2 % (p > 0.05) of the photosynthetic carbon as isoprene, respectively. With increasing temperatures, heat-stressed trees emitted an equivalent of up to 12 % (temperature > 45°C) and heat–drought stressed trees up to 20 % of the assimilated carbon as isoprene (temperature range of 40 – 45°C).

**3.4 Changes in light and temperature curves of isoprene emission during stress**

The light and temperature relationships of isoprene emissions were parameterized for all treatments including only measurements taken during the period of the heat waves (Table 3). Details for all parameters optimized by recursively fitting the nonlinear light and temperature equations (Eq. 7 and 8) to the bin-averaged isoprene data are given in Table 3. Except for the parameter $C_{T1}$ in the heat–drought treatment all fitted values were statistically significant.

When comparing isoprene emissions of the stress treatments with the control, isoprene emissions at light saturation (and 30 °C), were approximately 45 % lower in the heat and 24 % lower in the heat–drought treatment (Fig 5b). Compared to literature values isoprene emissions of all trees reached light saturation at relatively low values of photosynthetically active radiation (e.g. PAR between 200 and 300 µmol m$^{-2}$s$^{-1}$ for the control and heat–drought stressed trees), most probably because of the relatively low levels of PAR (see Table 1) in the greenhouse. Response curves of isoprene versus environmental conditions obtained in this study will thus be more comparable to lower canopy conditions where leaves are not constantly under light saturation. To reflect this, we normalized temperature responses of isoprene to a PAR of 500 µmol m$^{-2}$s$^{-1}$ (typically values of

1000 μmol m$^{-2}$s$^{-1}$ are used in the literature). Normalized temperature response curves revealed, similar to light response curves, that control trees emitted (in the temperature range that overlapped among all treatments) more isoprene than stressed trees at similar temperatures (Fig. 5a). The 95 % confidence bounds for the fitted curves (dashed lines) indicate that above 30°C temperature functions of the heat and heat–drought trees were statistically different to the temperature function of the control trees. At a standard temperature of 30°C, for example, the control trees emitted $16.0 \pm 0.2$ nmol m$^{-2}$s$^{-1}$ of isoprene, while the heat-stressed trees and heat–drought stressed trees emitted $8.7 \pm 0.5$ nmol m$^{-2}$s$^{-1}$ and $12.1 \pm 1.2$ nmol m$^{-2}$s$^{-1}$, respectively (see also Table 4). However, not only the emission factor $E_S$ but also other parameters ($C_{T1}$, $C_{T2}$ and $T_m$) related to the shape of the temperature response function changed during heat and heat-drought stress (see Table 4). For temperatures above 28°C, average values of isoprene emissions differed significantly among the control and stress treatments (Appendix B, Table B1).

**3.5 Modeled leaf-level isoprene emissions**

Measured isoprene emissions were assessed against modeled values (for model parameters see Table 4). As expected the heat stress (slope = 0.95 and $R^2 = 0.89$) and control models (slope = 1.01 and $R^2 = 0.87$) performed quite well in estimating the isoprene emissions over the whole temperature range covered by the measurements. The model derived for combined heat-drought stress was, however, less successful in estimating measured isoprene emissions (slope = 0.73 and $R^2 = 0.65$) and tended to underestimate high isoprene emission rates (Fig. 6).

To assess the possibility of simulating isoprene emissions of heat or heat-drought treated trees based on the parameters from the control treatment, we derived a correction factor from the difference of isoprene emission rates from treatment-based model parameterization versus the control-based model (Fig. 7a). The derived correction factors are similar for both treatments (0.47 for the heat treatment and 0.49 for the heat-drought treatment; Fig. 7a). However, when applying the correction factor, the thereby modified models failed to simulate the measured isoprene emissions for heat and heat-drought trees (Fig. 7b), in particular in the higher flux range. This indicates that a simple linear adjustment represents a poor substitute for a stress-specific parameterization to simulate isoprene emissions.

## 4 Discussion

### 4.1 Stress response and recovery

In our study, heat and heat–drought stressed trees showed reduced stomatal conductance along with lower rates of net photosynthesis during the heat waves (Fig. 3). While net photosynthesis in black locust is limited by stomatal closure during heat and heat–drought stress (Ruehr et al., 2016), isoprene emission is mostly insensitive to the degree of stomatal opening (i.e., high Henry´s law constant, see Niinemets et al., 2004). Additionally the temperature optimum of net photosynthesis is usually reached at much lower temperatures than that for isoprene synthase activity (Rennenberg et al., 2006), resulting in an earlier inhibition of photosynthesis compared to isoprene emissions (Loreto and Fineschi, 2015). In our experiment, heat and heat–drought stressed black locust trees showed a temperature optimum of net photosynthesis at about 25°C, while peaks in isoprene emissions were reached at much higher temperatures (42.4°C in the heat and 41.2°C in the heat–drought treatment) similar to what has been reported for other tree species (Guenther et al., 1991, 1993; Monson et al., 1992). The temperature optima of isoprene synthase and other enzymes, as well as the availability of the isoprene precursor dimethylallylpyrophosphate, are likely responsible for this threshold (Niinemets et al., 2010a) and we can expect isoprene emissions to increase unless these temperature optima are reached or carbon substrate for isoprene synthase has become depleted (Grote and Niinemets, 2008). In agreement with our study, earlier studies on heat stress responses found elevated isoprene emissions (Sharkey and Loreto, 1993; Singsaas and Sharkey, 2000). This is in stark contrast to patterns of isoprene emissions during drought, where most studies found no change or even reduced emissions (Pegoraro 2004a; Brilli et al., 2007; Fortunati et al., 2008). In our study, however, the effects of drought were apparently dominated by the responses of isoprene emissions to the high temperatures as both, heat and heat–drought stressed trees showed similar emissions. This may indicate that hotter droughts as predicted with climate change could lead to enhanced leaf-level isoprene emissions in black locust.

Upon stress release, isoprene emissions recovered more quickly (within about two days) than net photosynthesis to pre-stress levels. After periods of drought stress, a quick recovery of isoprene emissions seems to emerge as a common feature that has also been observed in previous studies (Sharkey and Loreto, 1993; Pegoraro et al., 2004b; Velikova and Loreto, 2005; Brilli et al., 2013) and may help isoprene emitting plants to cope with abrupt and repeated temperature changes as commonly observed under natural conditions. However, studies on isoprene dynamics following stress release are scarce and there is to our knowledge only one study that considers dynamics of isoprene emissions during and following prolonged combined heat–drought stress (Vanzo et al., 2015). Vanzo et al. (2015) found that isoprene-emitting poplars recovered rapidly from stress and even increased photosynthesis during recovery in contrast to non-isoprene-emitting trees, which showed weaker recovery of photosynthesis. Net photosynthesis in our study recovered quickly, too. Comparing isoprene and non-isoprene emitting poplars, Vanzo et al. (2015) could demonstrate a positive effect of isoprene emissions on the trees' performance during and following heat-drought stress under ambient $CO_2$. Such a beneficial effect of isoprene during high temperature stress and a quick recovery thereafter has also been reported by other studies (Velikova and Loreto, 2005, Behnke et al., 2007). Thus, the fast recovery of both isoprene emission (reduction) and photosynthesis (increase) could be related to the beneficial effects of isoprene synthase in protecting the photosynthetic apparatus. Further, a fast recovery of photosynthesis and isoprene emissions after stress, suggests

that no irreversible damage to the unshed leaf tissues in consequence of high temperature or drought had been induced (Niinemets, 2010).


## 4.2 Isoprene emissions and photosynthetic carbon gain

Photosynthesis supplies most of the carbon, as well as energy for isoprene synthase during unstressed conditions (Delwiche and Sharkey 1993; Sharkey and Loreto, 1993; Niinemets et al., 1999; Karl et al., 2002). Assuming that all the carbon incorporated into isoprene originates directly from photosynthesis, approximately 1.6 % of the
carbon assimilated during net photosynthesis was used for isoprene emission in control trees at temperatures between 28°C and 32°C. This value is in the same range as the 2 % which were proposed for other major isoprene emitting trees at a temperature of 30°C (Sharkey and Yeh, 2001; Sharkey et al., 2008). The ratio of photosynthesis to isoprene emission can however change dramatically during stress conditions. In a coppice poplar plantation at ambient temperatures only 0.7 % of assimilated carbon were emitted as isoprene (Brilli et
al., 2016), while in response to high temperature or drought stress, the ratio of isoprene emission to assimilated C may increase up to 50 % or even more (Sharkey and Loreto, 1993; Pegoraro et al., 2004b). In our experiment we found a decoupling of isoprene emissions from photosynthesis and thus the equivalent of up to 13 % of C assimilated in the heat and 20 % in the heat–drought treatment to be emitted as isoprene (Table 3). The divergence between isoprene emissions and photosynthesis is most likely a consequence of the different
temperature optima for isoprene emissions and photosynthesis. While above 30°C isoprene emissions are still increasing exponentially with temperature, photosynthesis is already decreasing (Ruehr et al. 2016) leading to the discrepancies between photosynthesis and isoprene emissions. Although we assume that isoprene is mainly formed from current photosynthates, we cannot exclude that C for isoprene formation might have originated from other carbon sources such as sugars and starches (Affek and Yakir, 2003). Especially under conditions of
limited photosynthesis, like severe drought, it has been reported that plants use increasing amounts of stored C to supply isoprene synthesis (Brilli et al., 2007; Fortunati et al., 2008). The divergence between photosynthesis and isoprene emissions during stress as found in our study could indicate that re-mobilized C might have been used to supply isoprene synthesis, originating from non-structural carbohydrates in leaves or other tissues (Schnitzler et al., 2004).

It is still a matter of debate why some plants invest substantial amounts of carbon to maintain isoprene emissions even under severe stress when C demand for maintenance might be higher than C supply. One likely explanation is that isoprene acts as antioxidant in the plants eliminating reactive oxygen species produced during stress in order to prevent oxidative damage (Vickers et al., 2009). Further, isoprene is discussed to protect the chloroplasts under high temperatures or drought (Velikova et al., 2011; Velikova et al., 2016) which was
explained with a stabilizing effect of isoprene on the thylakoid membranes (Velikova et al., 2011). This in turn has been again reported to reduce the formation of reactive oxygen species (Velikova et al., 2012). However, Harvey et al. (2015) found that the concentration of isoprene within the leaves is lower than expected and thus unlikely to alter the physical properties of the thylakoid membranes. Thus, the exact pathways leading to the thermoprotective effect of isoprene are still in discussion.


### 4.3 Changes of isoprene temperature and light response functions during stress

Common knowledge about the temperature response function of isoprene (Niinemets et al., 2010) would suggest that the higher isoprene emissions for stressed plants found here are solely due to increased temperatures. However, heat and heat–drought stressed trees showed somewhat different temperature and light response curves and had 45 % and 25 % lower isoprene emissions relative to the control trees at the chosen standard temperature of 30°C. If isoprene emissions of the stressed trees are calculated with the parameter values of the control trees, the average isoprene emissions during stress would have been overestimated by roughly 50 % in both stress treatments. At a first glance, the apparent lower isoprene emissions in stressed trees compared to unstressed trees at the same temperature is surprising. Intuitively one would expect heat and heat–drought stressed plants to emit – in comparison to control trees at the same temperature – more isoprene during periods of stress because of the thermoprotective role of isoprene (Vickers et al., 2009; Ryan et al., 2014). However, there are indications that the quantities of emissions have little influence on the thermoprotective effect, as long as isoprene emissions are maintained. Vickers et al. (2009) stated that even low isoprene emissions should be sufficient for the stabilization of membranes under heat stress. A fact which supports this theory is that external isoprene fumigation of non-isoprene-emitting plants increases their thermotolerance (Velikova and Loreto, 2005), while it has no effect in in isoprene-emitting species (Logan and Monson, 1999).

The overestimation of isoprene emission rates simulated by the Guenther et al. algorithm (control treatment parameterization) in our study during prolonged stress episodes agrees with observations at the ecosystem scale: Potosnak et al. (2013) and Seco et al. (2015) found that isoprene emissions from an oak and hickory dominated deciduous forest and a broadleaf temperate forest were overestimated by the Guenther et al. algorithm during severe drought events. Brilli et al. (2016) reported that during a high temperature period, isoprene emissions of a poplar plantation simulated with the Guenther et al. algorithm were higher than observed emissions. A reduction of isoprene emission rates under prolonged, but moderate stress does not only hold for drought stress, but has been found for several abiotic stressors (Niinemets, 2010). It is also known that the severity and duration of stress plays a crucial role in the actual stress response especially in case of irreversible damage (Niinemets, 2010).

### 4.3 Modelling isoprene emissions during stress

While the basic shapes of temperature and light response functions of isoprene emissions have been manifested quite early (Guenther et al., 1993), recent research reports that these response functions can vary with previous environmental conditions (Niinemets et al., 2010b) and may critically depend on the experimental conditions (e.g. how long isoprene emissions were allowed to equilibrate after changing the temperature). The standard isoprene emission factor is not necessarily a species-specific constant, but may change in response to stress, leaf age, or $CO_2$ concentration (Niinemets et al., 2010a). Thus we assessed the response of isoprene emissions to changes in temperature and light. In contrast to most studies this was not done on the basis of discrete changes in temperature and light but under fluctuating environmental conditions (following outside temperature and humidity), which may naturally occur. We were able to simulate measured isoprene emissions using treatment-specific model parameters (Table 4) with large confidence ($R^2 > 0.87$ and slope close to one), only under heat-

drought conditions highest isoprene emissions were underestimated by the treatment-specific model. In summary, our finding that the isoprene emission factor $E_S$ reduces during heat and heat-drought stress, agrees well with current findings (Niinemets et al., 2010a; Geron et al., 2016). However, by applying a simple correction factor the standard-parameterized (control) model did not allow simulating the stress-induced changes in emissions. This is because not only the emission factor, but also the shape of the temperature response function changed with stress This expands findings, which have been reported recently by Geron et al. (2016) for several oak species during drought. However, while for most oak species the $E_S$ was, similar to our study, reduced during drought, in one out of five oak species isoprene emission rates increased (Geron et al., 2016). This shows on the one hand that such stress effects may be species-specific and on the other hand that neither a general direction of the change in $E_S$ nor the change in the shape of temperature or light response functions is known. Therefore, further field and laboratory studies will be required to reach a level of process understanding that allows describing stress-driven isoprene emissions. In any case, the results of this study clearly show the need to critically reassess temperature response functions during stress and, in a further step, incorporate stress-specific response functions into BVOC emission models in order to provide reliable estimates.

**5 Conclusion**

We assessed isoprene emission patterns of black locust trees during two stress scenarios which are likely to occur more often in the future: prolonged heat and combined heat and drought stress. We not only investigated how trees will respond to and recover from such events under close-to natural conditions, but also how heat and heat-drought stress alters temperature and light response functions of isoprene emissions, typically used to predict isoprene emissions. While overall isoprene emissions increased in response to higher temperatures, we found that this increase was lower than what would be predicted from temperature response curves of unstressed trees. In addition, we showed that a simple correction factor did not allow simulating stress-driven isoprene emissions, due to the non-linear nature of the stress-driven changes. For simulating isoprene emissions under periodic heat or combined heat-drought stress, it will thus be necessary to critically reassess the temperature and light response functions typically used. In the light of climate change revised stress response functions are important to allow future projections of BVOC emissions, including air quality and air chemistry predictions. Moreover, BVOC-specific stress response functions need to be developed considering their different physiological roles and effects on air chemistry.

### Appendix A: Leaf Temperature

Since air temperature in the leaf-chambers was evaluated instead of leaf temperature we performed some additional leaf measurements during the second heat wave using an infrared camera (PI450 Optris GmbH, Berlin, Germany) to evaluate if leaf temperature of the trees did significantly differ from air temperature. However, the measurements suggested that independent of the treatment, leaf temperature was statistically (pairwise t-test with $p > 0.05$) not different from air temperature (Table A1).

### Appendix B: Bin-averaged Isoprene Emissions for Different Temperatures

To test if bin averaged isoprene emissions for different temperature classes did differ among treatments we performed a t-test which confirmed that for temperatures $> 28°C$ the isoprene emission of stressed trees was statistically different from the isoprene emission of control trees (Table B1).

**Acknowledgements** We would like to thank the IMK-IFU IT and mechatronics team, especially Christoph Soergel, Stefan Schmid and Bernhard Thom for their support and Martina Bauerfeind for her lab assistance. Further we would like to acknowledge Jukka Pumpanen from the University of Eastern Finland, Kuopio for his advice with the chamber design and setup.

This research was supported by the German Federal Ministry of Education and Research (BMBF), through the Helmholtz Association and its research program (ATMO), and a grant to Almut Arneth from the Helmholtz Association Innovation and Networking fund. I. B. and N.K.R. acknowledge support by the German Research Foundation through its Emmy Noether Program (RU 1657/2-1).

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

**Table 1:** Before stress growth conditions for black locust. Average $CO_2$ concentration, temperature, relative humidity (RH), and daytime photosynthetically active radiation (PAR>100 µmol m$^{-2}$s$^{-1}$) including the corresponding standard deviation in the two greenhouse compartments between 7 May and 13 June 2014, before the start of the first heat wave. Difference in growth conditions between the two greenhouse compartments are given in percent.

| growth conditions 07.05.14 – 13.06.14 | compartment 1 | | compartment 2 | | |
|---|---|---|---|---|---|
| | average | standard deviation | average | standard deviation | difference (%) |
| $CO_2$ (ppm) | 409 | 39 | 404 | 36 | 1.2 |
| Temperature (°C) | 15.6 | 5.4 | 15.6 | 5.2 | 0 |
| RH (%) | 80.8 | 13.8 | 82.1 | 12.8 | 1.6 |
| daytime PAR (µmol m$^{-2}$s$^{-1}$) | 419 | 286 | 412 | 248 | 1.9 |


**Table 2:** Results of a linear mixed-effects model evaluating isoprene emissions and daytime photosynthesis under different treatments and for different time-periods during the experiment (pre-treatment, heat period 1, recovery1, heat period 2 and recovery 2). The model tests for interactions between treatment and time-period relative to control conditions (ns means not significant, *** corresponds to a p-value<0.005 and (*) corresponds to a p-value between 0.05 and 0.15).

| | Isoprene | | | |
|---|---|---|---|---|
| | value | SE | t-statistics | significance |
| Pre-treatment: Control (Intercept) | 0.8 | 1.3 | 0.6 | ns |
| Pre-treatment: Heat | 0.1 | 1.2 | 0.1 | ns |
| Pre-treatment: Heat–drought | 1.4 | 1.2 | 1.1 | ns |
| Stress period 1 | 1.2 | 1.3 | 0.9 | ns |
| Recovery 1 | 0.2 | 1.5 | 0.1 | ns |
| Stress period 2 | 0.2 | 1.3 | 0.1 | ns |
| Recovery 2 | -0.4 | 1.5 | -0.3 | ns |
| Heat x stress period 1 | **4.4** | **1.1** | **3.8** | *** |
| Heat–drought x stress period 1 | **3.4** | **1.2** | **3.0** | *** |
| Heat x Recovery 1 | 0.2 | 1.2 | 0.1 | ns |
| Heat–drought x recovery 1 | -0.6 | 1.2 | -0.5 | ns |
| Heat x stress period 2 | **1.8** | **1.1** | **1.6** | (*) |
| Heat–drought x stress period 2 | **4.0** | **1.1** | **3.6** | *** |
| Heat x recovery 2 | -0.2 | 1.2 | -0.1 | ns |
| Heat & drought x recovery 2 | -1.3 | 1.2 | -1.1 | ns |
| | Photosynthesis | | | |
| | value | SE | t-statistics | significance |
| Pre-treatment: Control (Intercept) | 4.6 | 0.8 | 5.9 | *** |
| Pre-treatment: Heat | 0.0 | 0.8 | 0.0 | ns |
| Pre-treatment: Heat–drought | -0.8 | 0.8 | -1.1 | ns |
| Stress period 1 | 0.0 | 1.2 | 0.0 | ns |
| Recovery 1 | 0.8 | 1.6 | 0.5 | ns |
| Stress period 2 | 1.0 | 1.2 | 0.8 | ns |
| Recovery 2 | 1.0 | 1.9 | 0.5 | ns |
| Heat x stress period 1 | **-1.9** | **0.7** | **-2.8** | *** |
| Heat–drought x stress period 1 | **-1.9** | **0.7** | **-2.8** | *** |
| Heat x Recovery 1 | -1.1 | 0.9 | -1.2 | ns |
| Heat–drought x recovery 1 | 0.4 | 0.9 | 0.4 | ns |
| Heat x stress period 2 | **-2.6** | **0.6** | **4.1** | *** |
| Heat–drought x stress period 2 | **-1.9** | **0.6** | **3.0** | *** |
| Heat x recovery 2 | **-1.7** | **1.1** | **-1.5** | (*) |
| Heat & drought x recovery 2 | -0.4 | 1.0 | -0.4 | ns |

**Table 3:** Ratio between assimilated carbon and carbon emitted as isoprene ($C_{iso}/C_A$) averaged for different temperature ranges and treatments during the stress periods including the corresponding standard deviation (calculated using data points with PAR > 50 µmol m$^{-2}$s$^{-1}$ only). The number of values (n) included in the calculation of the class averages is given to the right of each $C_{iso}/C_A$ column. $C_{iso}/C_A$ values were only significantly different (p<0.05, u-test) between the control and heat treatment in the temperature range 28°C-32°C.

| Treatment | Control | | Heat | | Heat–drought | |
|---|---|---|---|---|---|---|
| Temperature range (° C) | $C_{iso}$ / $C_A$ (%) | n | $C_{iso}$ / $C_A$ (%) | n | $C_{iso}$ / $C_A$ (%) | n |
| 15-20 | 0.1±0.03 | 2 | --- | 0 | --- | 0 |
| 20-24 | 0.3±0.1 | 16 | --- | 0 | --- | 0 |
| 24-28 | 0.5±0.2 | 15 | 0.6* | 1 | 0.8±0.3 | 2 |
| 28-32 | 1.6±0.7 | 12 | 0.8±0.3 | 15 | 1.2±0.7 | 29 |
| 32-35 | --- | 0 | 1.7±0.7 | 29 | 3.0±2.5 | 32 |
| 35-40 | 6.5* | 1 | 5.3±6.0 | 38 | 10.0±16.4 | 49 |
| 40-45 | --- | 0 | 10.9±6.4 | 17 | 20.2±16.5 | 25 |
| >45 | --- | 0 | 12.5* | 1 | 12.0±5.4 | 5 |

* single value

**Table 4:** Parameters $E_S*C_{L1}$ ($E_S$ being the isoprene emission factor at standardized conditions and $C_{L1}$ a dimensionless scaling parameter) and α including their corresponding standard errors and the t-statistic for the optimized light response curve of the control heat and heat–drought trees at a standard temperature of 30°C. The parameters $E_S$ [nmol m$^{-2}$s$^{-1}$], $C_{T1}$ [J mol$^{-1}$], $C_{T2}$ [J mol$^{-1}$] and the temperature optimum of isoprene emissions $T_m$ [K] (with corresponding standard errors SE and t-statistic) derived for the temperature response curve at a standard photosynthetically active radiation of 500 µmol m$^{-2}$ s$^{-1}$ are shown in an analog manner. Values with a p-value ≤ 0.05 are given in bold.

| | | Light response curve | | Temperature response curve | | | |
|---|---|---|---|---|---|---|---|
| | | $E_S*C_{L1}$ | α | $E_S$ | $C_{T1}$ | $C_{T2}$ | $T_m$ |
| Control | value | **16.2** | **0.0070** | **16.0** | **1.42*10$^5$** | **5.08*10$^5$** | 311.80 |
| | SE | 1.4 | 0.002 | 0.2 | 0.05*10$^5$ | 0.83*10$^5$ | n.v. |
| | t-statistic | 11.5 | 4.1 | 74.8 | 29.2 | 6.1 | n.v. |
| Heat | value | **9.5** | **0.0043** | **8.7** | **1.38*10$^5$** | **2.83*10$^5$** | **315.5** |
| | SE | 0.4 | 0.0004 | 0.5 | 0.13*10$^5$ | 0.25*10$^5$ | 1.0 |
| | t-statistic | 22.5 | 11.2 | 17.7 | 10.7 | 11.5 | 318.8 |
| Heat - drought | value | **12.8** | **0.0037** | **12.1** | 1.01*10$^5$ | **2.80*10$^5$** | **314.3** |
| | SE | 0.7 | 0.0004 | 1.2 | 0.25*10$^5$ | 0.46*10$^5$ | 2.0 |
| | t-statistic | 17.7 | 8.4 | 10.2 | 4.0 | 6.1 | 155.2 |

**Table A1**: Air temperature ($T_{air}$) and corresponding leaf temperature ($T_{leaf}$) including standard errors are given for the control, heat, and heat-drought treatment. Leaf temperature was measured with an infrared camera on two days during the second heat wave. Differences between leaf and air temperature were not significant (pairwise *t*-test *p* >0.05).

| Treatment | $T_{air} \pm$ SE (°C) | $T_{leaf} \pm$ SE (°C) | $T_{leaf}$ - $T_{air}$ (°C) | p < 0.05 | n |
|---|---|---|---|---|---|
| **Control** | 21.8 ± 1.9 | 21.1 ± 1.9 | 0.7 ± 0.5 | n.s. | 3 |
| **Heat** | 34.7 ± 1.2 | 34.4 ± 1.5 | -0.3 ± 0.5 | n.s. | 5 |
| **Heat-drought** | 36.0 ± 0.5 | 35.2 ± 0.8 | -0.9 ± 0.5 | n.s. | 10 |

**Table B1**: Bin-averaged isoprene emissions ($E_{iso}$) for different temperature classes including the corresponding standard errors (if n>1). Significant differences in $E_{iso}$ between treatments and control are given in bold.(p<0.05 based on a t-test if the number of measurements exceeded three). Values highlighted by an asterisk indicate significant differences between the heat and heat-drought treatment.

| Treatment | Control | | Heat | | Heat-drought | |
|---|---|---|---|---|---|---|
| Temperature range (°C) | $E_{iso}$ | SE | $E_{iso}$ | SE | $E_{iso}$ | SE |
| | (nmol m$^{-2}$s$^{-1}$) | | (nmol m$^{-2}$s$^{-1}$) | | (nmol m$^{-2}$s$^{-1}$) | |
| 15-20 | 1.52 | 0.01 | - | - | - | - |
| 20-24 | 3.81 | 0.42 | - | - | - | - |
| 24-28 | 7.33 | 0.76 | 4.95 | 1.23 | 9.83 | 2.31 |
| 28-32 | 16.14 | 1.81 | **9.13** | **0.81** | **10.50** | **1.17** |
| 32-35 | 23.12 | - | 16.62 | 0.82 | 18.83 | 1.82 |
| 35-40 | 40.41 | - | 26.26 | 1.03 | 24.86 | 1.83 |
| 40-45 | - | - | **37.43\*** | **2.58** | **23.36\*** | **2.60** |
| >45 | - | - | 32.35 | - | 13.54 | 4.24 |

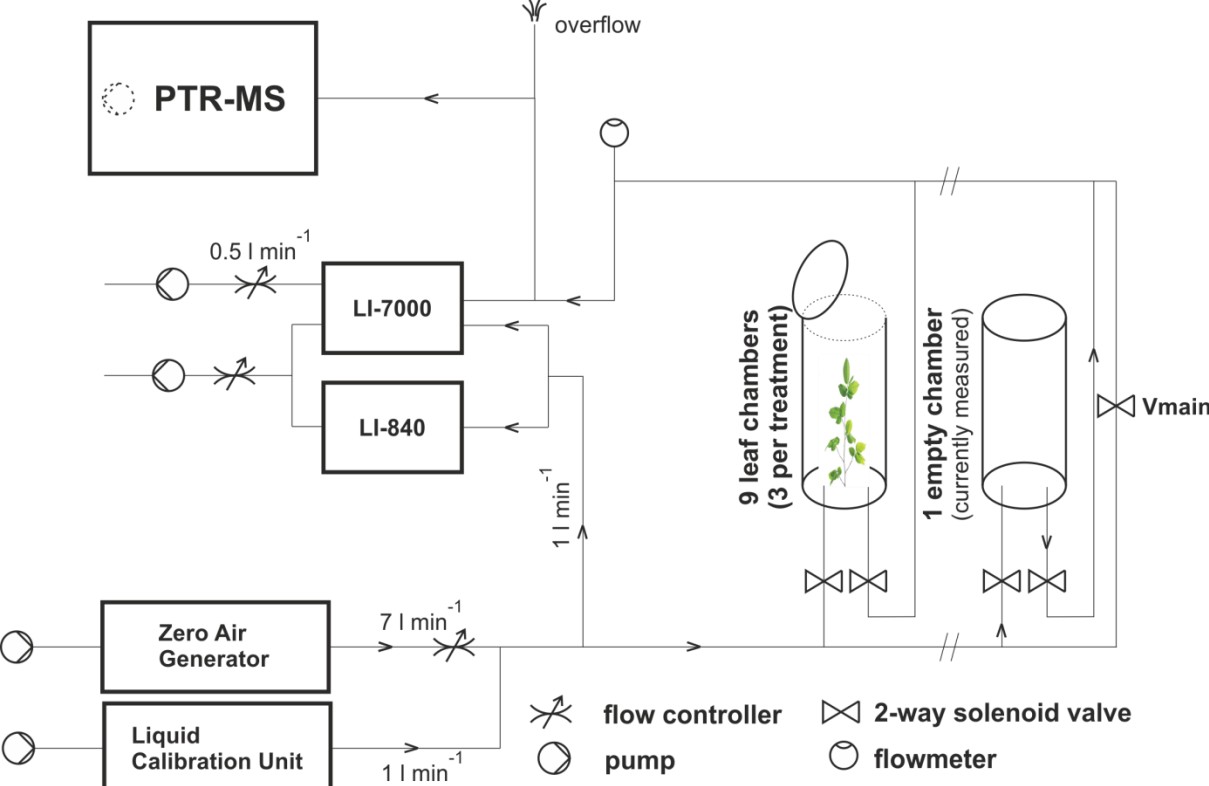

**Figure 1:** Schematic of the automated gas exchange measurement set-up. Note that for simplification the setup is shown for two chambers, but was extended to 9 leaf chambers and one empty chamber measured in sequence. Leaf chambers were made of cylindrical Plexiglas coated with a thin Teflon layer. Leaf chambers remained open all times, except during measurements when a movable lid was automatically closed (see further details in Methods section). The direction of the air flow is indicated by the small arrows.


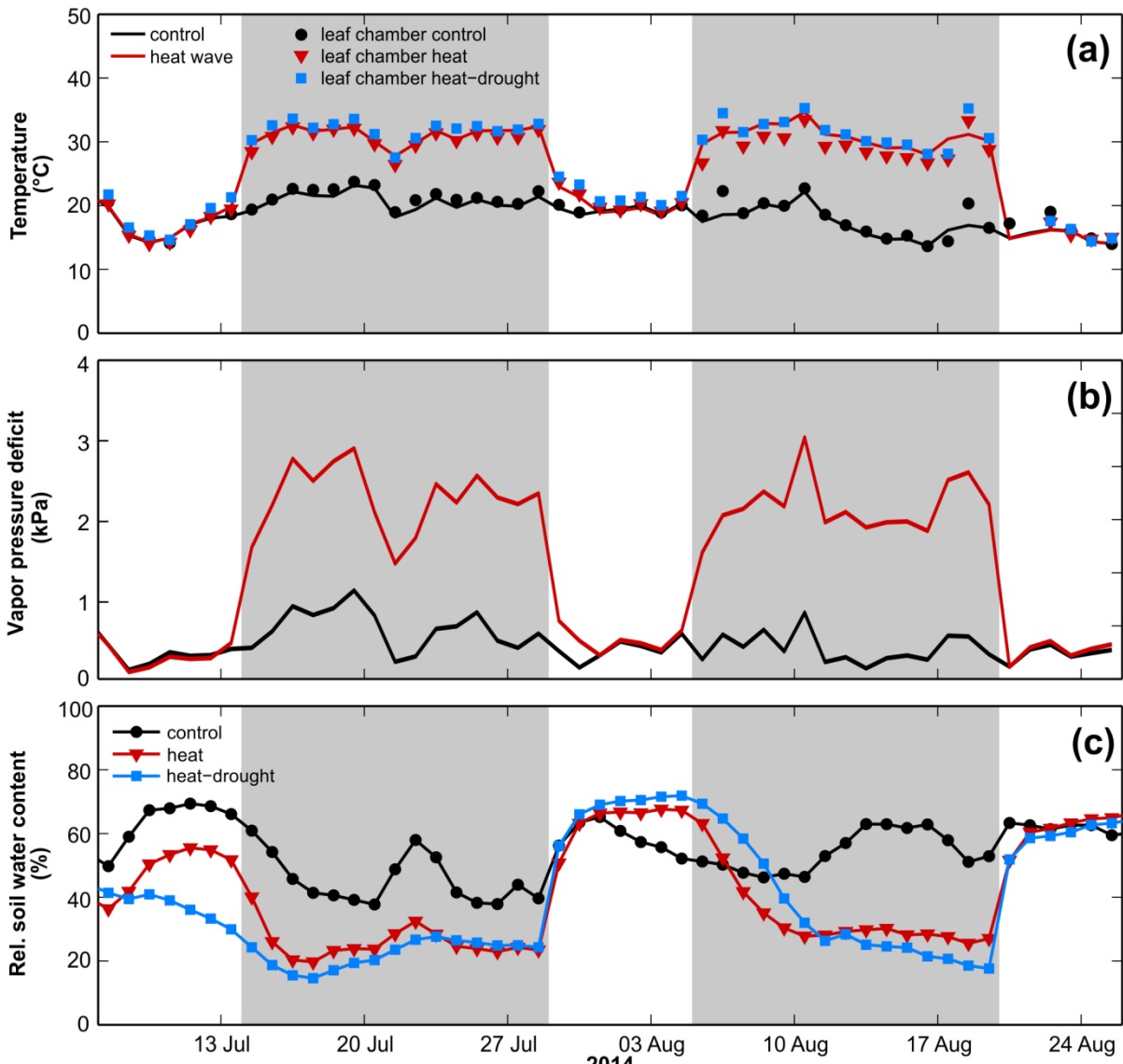

**Figure 2:** Daily average temperatures (a) in the control (black line) and stress (red line) compartment of the greenhouse and in the plant chambers of the control (black circles), heat (red triangles) and heat–drought (blue squares) treatments. Daily average vapor pressure deficit (b) in the control (black line) and heat (red line) compartment of the greenhouse and relative soil water content (c) averaged for each treatment (control – black line and symbol, heat – red line and symbol, heat–drought – blue line and symbol) and measurement day. Heat waves are represented by the grey colored areas.

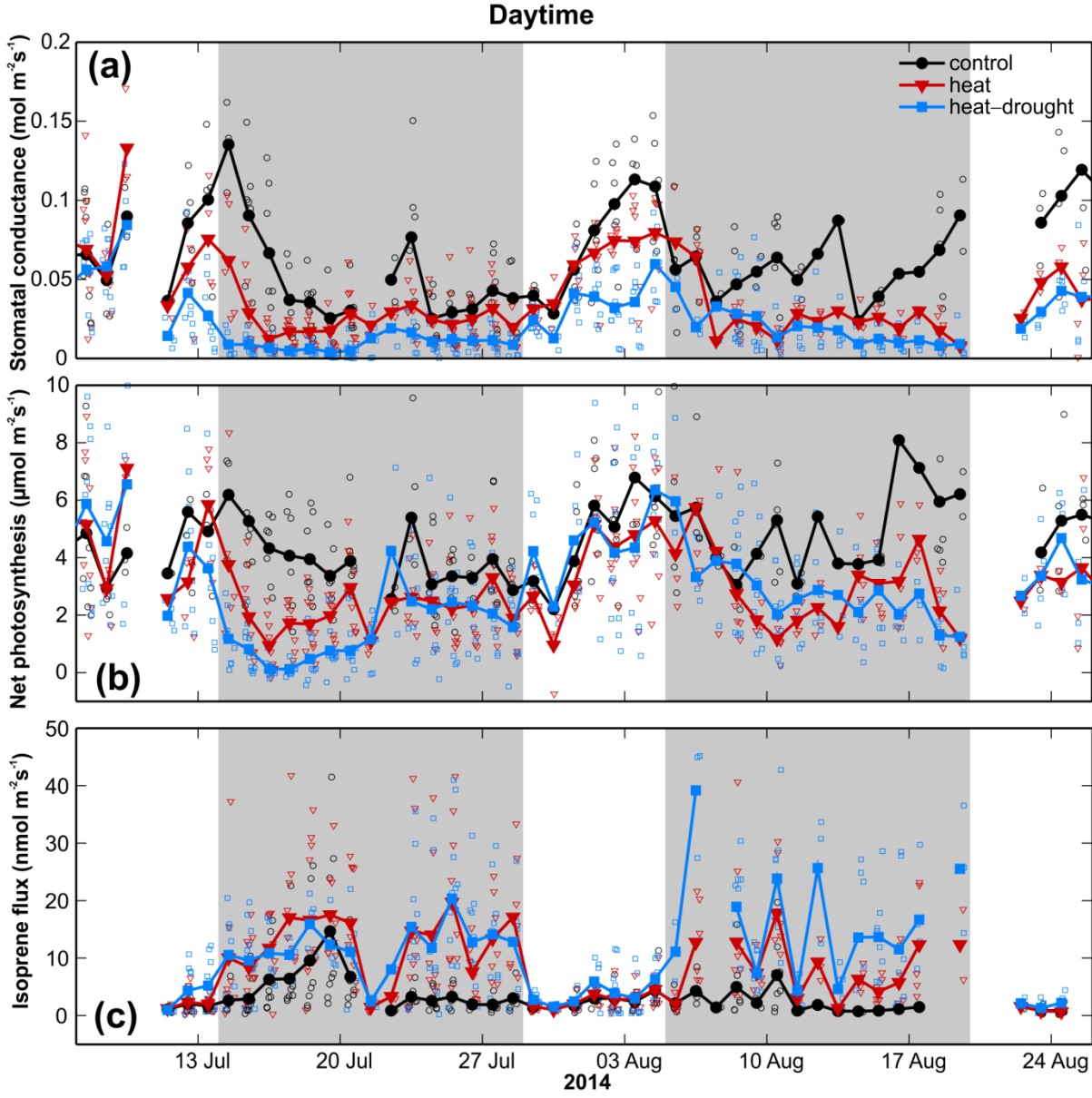

**Figure 3:** Daytime (PAR > 50 µmol m$^{-2}$s$^{-1}$) values for stomatal conductance (a) photosynthesis (b) and isoprene emission (c) of black locust trees for the control (black circles), heat (red triangles) and combined heat–drought treatment (blue squares). Filled symbols and lines are daytime averages on average consisting of seven single chamber measurements. Heat waves are represented by the grey colored areas.


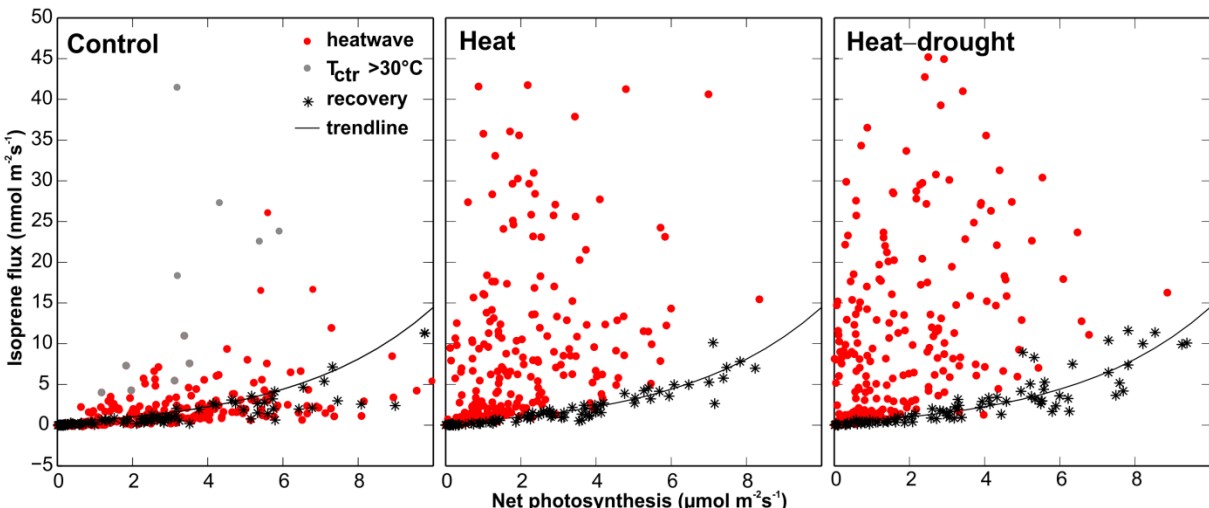

**Figure 4:** Relationship of isoprene emission with photosynthesis ($> 0$ µmol m$^{-2}$s$^{-1}$) in black locust trees during the two heat waves (red and grey circles; grey circles distinguish points when the temperature in the control chambers exceeded 30°C) and recovery periods (black asterisks) shown in separate panels for the control, heat and heat–drought treatment. Solid lines represent an exponential curve of the form y=exp$^{(\alpha\ x)}$-β which was derived from a non-linear fit to the measurements of heat–drought stressed trees during recovery to describe the

dependency between photosynthesis and isoprene emission exemplarily.

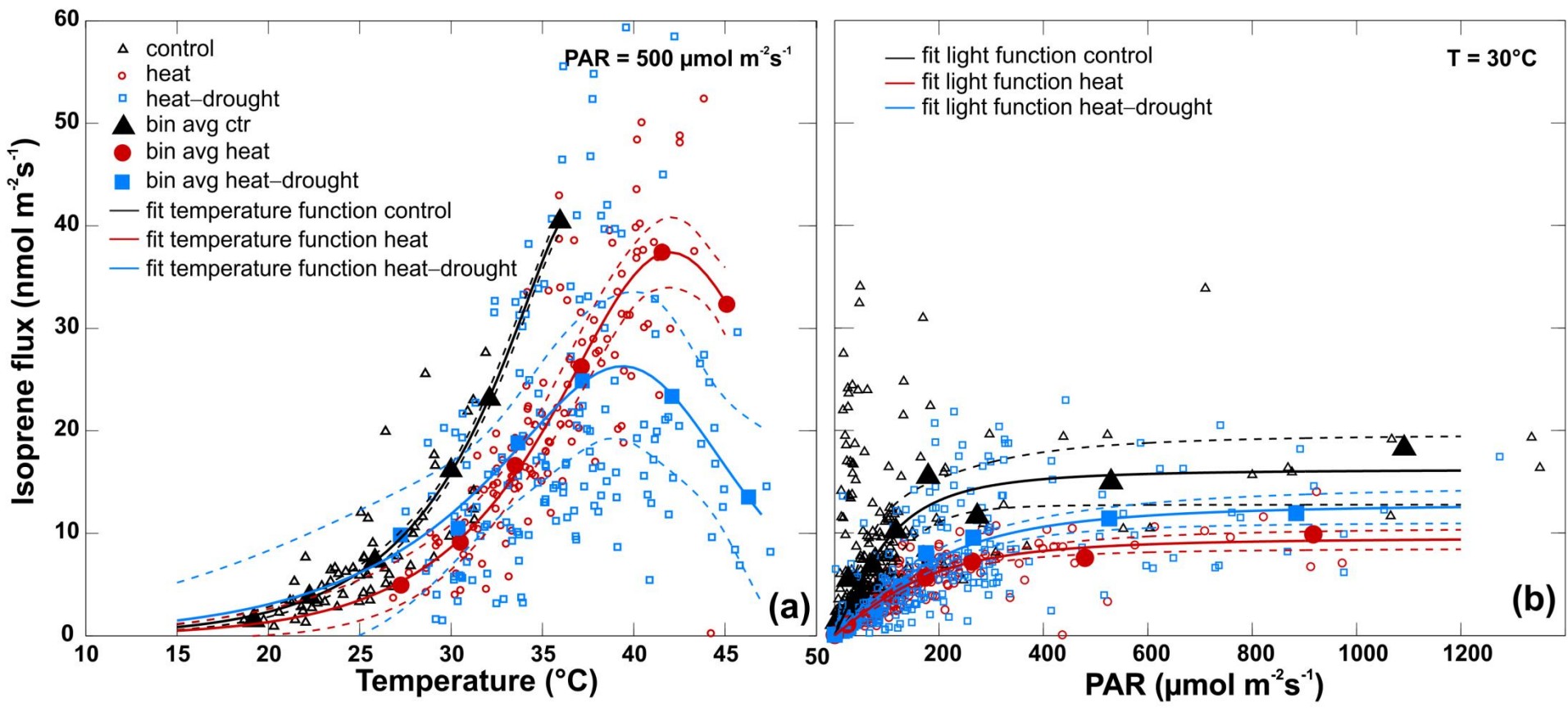

**Figure 5:** Dependency of isoprene emissions on temperature (a; isoprene emissions for PAR > 100 µmol m$^{-2}$s$^{-1}$, normalized to PAR = 500 µmol m$^{-2}$s$^{-1}$) and light (b; normalized to T = 30°C) in black locust trees during the heat waves. Filled symbols are bin averages for predefined temperature or light classes in the control (black), heat (red) and heat–drought treatment (blue). Single data points are depicted by open symbols. Solid lines are derived by non-linear regression of averaged isoprene emissions to temperature and light response functions (Eq. 7 and 8). Dashed lines are the respective 95 % confidence intervals of the regression fits. For model parameters see Table 4.

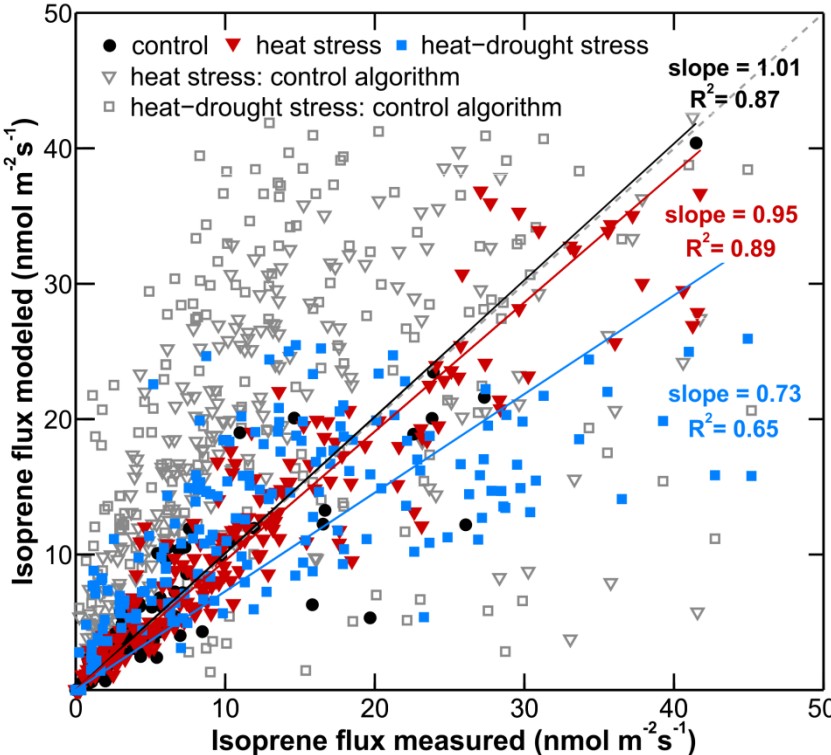

**Figure 6:** Modelled versus measured isoprene fluxes for trees exposed to control conditions (black circles), heat
stress (red triangles), and heat-drought stress (blue squares) including a linear least square fit. Open grey
symbols show isoprene fluxes modeled with the control algorithm instead of the corresponding algorithm for
heat and heat-drought stressed trees.

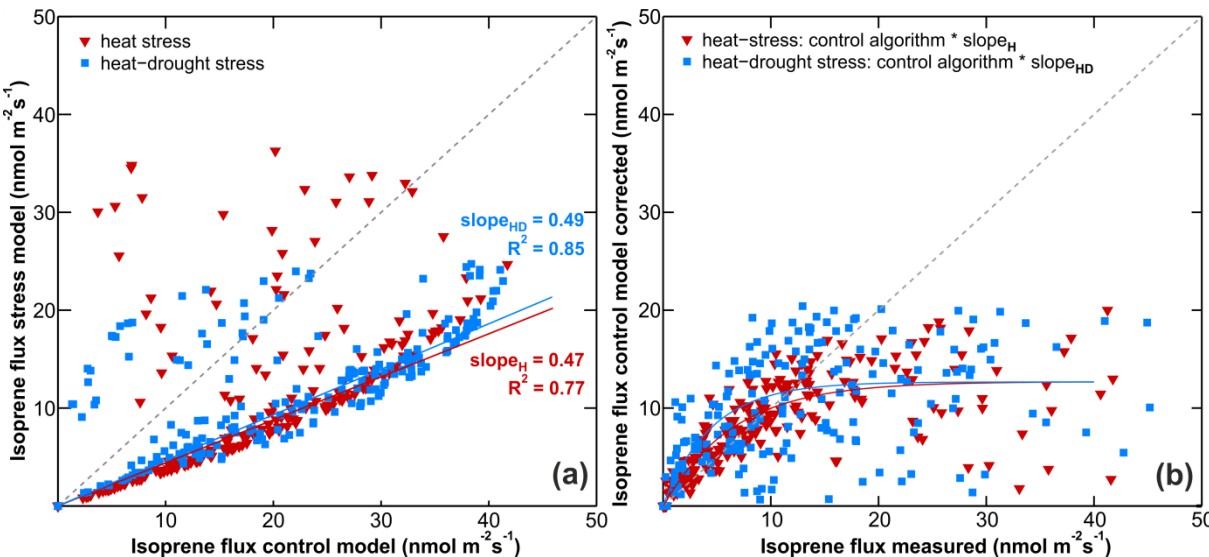


**Figure 7:** a) Isoprene fluxes of heat and heat-drought stressed trees modeled with the stress algorithm against fluxes modeled with the control algorithm including a linear least-square fit showing the slope which would bring fluxes calculated with the control algorithm in line with fluxes calculated with the stress algorithm; b) Isoprene fluxes modeled with the control algorithm and corrected with the slope denoted in S1a to account for

changes in the standard isoprene emission rate during stress.