# Peer review of "Isoprene emission and photosynthesis during heat waves and drought in black locust"

_Biogeosciences, 2017_

## Referee Comment (RC1) · Anonymous Referee #1 · 24 Feb 2017

Isoprene emissions in relationship to plant carbon cycling during drought and high temperature stress is an important and active area of research with numerous papers on this subject in recent years. Isoprene production protects carbon assimilation processes including stabilizing photosynthetic membranes during high temperature stress through numerous potential mechanisms including excess photosynthetic energy consumption, direct antioxidant activity, physical membrane stabilization, and signaling activities of oxidation products. The present study by Bamberger et al. investigated isoprene emissions and net photosynthesis responses in black locust trees growing under controlled environmental conditions before, during, and after drought and heat treatments. As observed in numerous other studies (e.g. Seco et al., 2015), net photosynthesis and isoprene emissions were coupled during non-stress conditions but became strongly uncoupled during heat and drought stress with substantial decreases

in net photosynthesis but a stimulation of isoprene emissions.

General Comments The paper generally lacks any new biochemical and physiological mechanistic description of how isoprene and net photosynthesis can become uncoupled at high temperatures and drought. Thus, it is not clear what new information the new study adds other than reporting these expected results in a new tree species. However, some novel aspects of the work include a characterization of the light and temperature responses of isoprene emissions during stress. However, the very low light saturation of isoprene emissions of both control and stressed trees (200-300 micromol/m2/s) indicates that the plants were not adapted to normal high light conditions of plants in natural ecosystems during the growing season). As only leaves from the lower canopy were measured, it is difficult to understand how these results can be used for modeling of natural isoprene emissions from nature. Studies show that the majority of photosynthesis and isoprene emissions from natural ecosystems occurs in the upper canopy leaves exposed to full sunlight.

Specific Comments Take care when refereeing to photosynthesis; the measurements are of net photosynthesis not of gross rates of photosynthesis, which can be drastically different under high temperatures.

PTR-MS signals at m/z 69 are not necessarily unique to isoprene, especially under drought or high temperature where C5 green leaf volatiles can significantly contribute to their signal (Fall et al. 2001). Since GC measurements were not performed, the results cannot be considered quantitative.

Suggested Citations

Seco, R., Karl, T., Guenther, A., Hosman, K. P., Pallardy, S. G., Gu, L., Geron, C., Harley, P. and Kim, S. (2015), Ecosystem-scale volatile organic compound fluxes during an extreme drought in a broadleaf temperate forest of the Missouri Ozarks (central USA). Glob Change Biol, 21: 3657–3674. doi:10.1111/gcb.12980

Fall R., Karl T., Jordon A. & Lindinger W. (2001) Biogenic C5VOCs: release from leaves after freeze-thaw wounding and occurrence in air at a high mountain observatory. Atmospheric Environment 35, 3905-3916.

---

## Referee Comment (RC2) · Anonymous Referee #2 · 27 Feb 2017

This manuscript makes the point that that isoprene emission model parameters are likely to be different for stressed plants than for unstressed plants. This is a good point to make. However, I found the manuscript to be problematic. Temperature of the individual leaves could not be controlled and so the temperature response curves of the control and heat or heat-drought treatments were almost non-overlapping. I found the description of the methods to be difficult. It is not clear to me whether leaves not currently being measured had an air flow of if the airflow only occurred during a measurement.

A great deal of variation in isoprene emission rates was observed. I was not convinced that the statistical treatments accurately reflected the variability. Isoprene emission is exceedingly difficult to predict, a point made by this lab that affects how these data need to be interpreted. While a lot of work has gone into this report, I have significant

concerns including that leaf temperature is not known and measurement temperature during isoprene emission measurement was almost non-overlapping.

The authors have an important point to make but the manuscript as written will not make that point very strongly. I made a number of comments on the pdf that I hope will be helpful to the authors.

Please also note the supplement to this comment:
http://www.biogeosciences-discuss.net/bg-2017-32/bg-2017-32-RC2-supplement.pdf

————————————————————

[Figure]

**Supplement:**

[revised manuscript text omitted]

---

## Referee Comment (RC3) · Anonymous Referee #3 · 28 Mar 2017

Summary:

This work examines the changes in foliar isoprene emission and photosynthesis of isoprene emitting black locust to periods of drought and drought plus heat stress. The isoprene and photosynthesis responses are compared to existing literature and the response of isoprene emission was then compared to that which would be calculated using the Guenther et al algorithm under the same stress conditions. In general the study is well written, although some aspects could be clearer (outlined in the comments sections below). There are a few typographical errors and the within text references need to be looked at as they should be presented either alphabetically or by date order. This study does appear to contain a solid body of work which is worth publishing to add to our understanding of isoprene responses to complex stresses and to help improve modelled emission estimates. However, as the manuscript is currently written I struggle

to find the novelty in the work. I have a few concerns and believe the manuscript should be improved as outlined below before it could be accepted for publication.

Major comments:

1) Materials and Methods, Experimental set up, line 96. I am concerned that the trees in the stress treatments had previously been exposed to two experimental heat waves and were showing a difference in basal area. Previous work has shown that VOC emissions differ based on exposure to previous environmental conditions (e.g. Sharkey et al , 1999 and citing references). Could the authors provide some reassurance that after pruning and over wintering the development and growth rates were then equivalent and could be fairly compared to one another? If they were not equivalent as suggested in the results section 3.1, were the data normalised?

2) Could the authors give an explanation as to why the trees were not randomly selected for the work included in the current study? This would have given a mixture of previously stressed and unstressed trees in each treatment group and removed any concern that the prior treatment of these trees was affecting the current results.

3) I would also like to see a clear description of the growth conditions and number of trees used per treatment and per measurement. Could the authors give a full description of the growth conditions of the trees (temperature, light, $CO_2$, RH) in the description of the experimental setup? Did the greenhouse have supplemental lighting, where was average PAR recorded, what was the day length? How many replicates were used per measurement? At the moment it is not clear to me how many replicates were used for what.

4) In general I cannot currently see the novelty of this work. However, this might be improved if the authors could use their data to suggest a new algorithm or an amendment to the existing algorithm to bring modelled isoprene emissions more in line with that which is observed. At the moment the authors highlight the difference between the observed and modelled emissions but don't go any further.

Minor comments

1) Abstract line 12 – mentions assessing the impact of stress on BVOC emissions but only isoprene is presented in the manuscript. Either remove the reference to general BVOC or include other emitted compounds.

2) Intro, line 38 – include ref to more recent Wyche et al, ACP 2014 which gives positive and negative effects of isoprene emission on secondary aerosol formation.

3) Into, line 65 and line 71 – include ref to more recent Ryan et al, New Phyt 2014 and remove older references unless they are seminal /original work.

4) Mat and Methods, Paragraph starting line 155 – description is not clear. Is the automatic switching of the measurements or the air flow? If air flow does this mean the chambers were clamped on the plants with no air flow for a period of time?

5) Section VOC Line 200 – the PTR-MS only counts set masses and cannot give compound identification. Could the authors include information on any mass identification that was performed (e.g. GC-MS) to confirm that it was only isoprene at m/z 69

6) Line 231 – 500 PAR seems quite low for trees in the summer. Top of canopy PAR in northern Europe during the summer is more likely to be between 1000 and 2000 PAR. Could the authors give a reason for choosing 500 PAR.

7) Mat & Methods Line 267 - Formatting error

8) Results 3.1 line 295. Could the authors include a description of how midday leaf water potential was measured?

9) Results 3.1 line 299 – typo "relative" should be "relatively"

10) Results 3.2 line 307 – I don't understand why " (PAR > 50 umol m-2 s-1)" is included in this sentence, when the sentence is referencing stomatal conductance – please clarify

11) Line 316 Daytime (PAR > 50 umol m-2 s-1) – I am assuming this means the authors collated any data collected when PAR readings were over this value to be "daytime" values. If this is correct please include a clarification at first use to make it easier for the reader to understand.

12) Line 322 – It may be over-stretching the results to include "marginally significant (p value around 0.1)" results as significant differences. This is not common practice but is perhaps personal preference.

13) Results 3.3, line 338 "significantly different to control trees" and "no significant differences. . ." please give p values.

14) Discussion Line 380 – references you should include more recent ref e.g. Ryan et al New Phyt 2014 who used genetically modified tobacco specifically to study the impact of drought on isoprene emission and protection.

15) Line 385 "A quick recovery of isoprene emissions after periods of drought stress seems to emerge as a 385 common feature that has also been observed in previous studies (Brilli et al., 2013; Pegoraro et al., 2004; Velikova and Loreto, 2005)" and line 288 "The observed faster recovery of isoprene emissions than photosynthesis may be a common pattern following stress release (Brilli et al., 2013; Pegoraro et al., 2004)." This appears to be a repeated point – please remove one of the sentences.

16) Line 390 – "this is the first study that considers dynamics of isoprene emissions during and following combined heat–drought stress. . ." Unfortunately this claim is untrue – please remove and see Vanzo et al, 2015 and references therein.

17) Paragraph beginning line 415 – including reference to Ryan et al, 2014, New Phytologist, who studied isoprene emitting and non-emitting plant responses to drought, would be appropriate here. Most likely with the Vickers et al, 2009 reference. 18) Table 2 – could the authors explain why there is such a variation in group sizes (n values from 0 – 49)?

[Figure]

References

Wyche, K. P., Ryan, A. C., Hewitt, C. N., Alfarra, M. R., McFiggans, G., Carr, T., Monks, P. S., Smallbone, K. L., Capes, G., Hamilton, J. F., Pugh, T. A. M., and MacKenzie, A. R.: Emissions of biogenic volatile organic compounds and subsequent photochemical production of secondary organic aerosol in mesocosm studies of temperate and tropical plant species, Atmos. Chem. Phys., 14, 12781-12801, doi:10.5194/acp-14-12781-2014, 2014.

Ryan, A. C., Hewitt, C. N., Possell, M., Vickers, C. E., Purnell, A., Mullineaux, P. M., Davies, W. J. and Dodd, I. C. (2014), Isoprene emission protects photosynthesis but reduces plant productivity during drought in transgenic tobacco (Nicotiana tabacum) plants. New Phytol, 201: 205–216. doi:10.1111/nph.12477

Sharkey, T., Singsaas, E., Lerdau, M., & Geron, C. (1999). Weather Effects on Isoprene Emission Capacity and Applications in Emissions Algorithms. Ecological Applications, 9(4), 1132-1137. doi:10.2307/2641383

Elisa Vanzo, Werner Jud, Ziru Li, Andreas Albert, Malgorzata A. Domagalska, Andrea Ghirardo, Bishu Niederbacher, Juliane Frenzel, Gerrit T.S. Beemster, Han Asard, Heinz Rennenberg, Thomas D. Sharkey, Armin Hansel, and Jörg-Peter Schnitzler (2015), Facing the Future: Effects of Short-Term Climate Extremes on Isoprene-Emitting and Nonemitting Poplar. Plant Physiol. 169: 560-575. First Published on July 10, 2015; doi:10.1104/pp.15.00871

---

## Author Comment (AC1) · 28 Apr 2017

Isoprene emissions in relationship to plant carbon cycling during drought and high temperature stress is an important and active area of research with numerous papers on this subject in recent years. Isoprene production protects carbon assimilation processes including stabilizing photosynthetic membranes during high temperature stress through numerous potential mechanisms including excess photosynthetic energy consumption, direct antioxidant activity, physical membrane stabilization, and signaling activities of oxidation products. The present study by Bamberger et al. investigated isoprene emissions and net photosynthesis responses in black locust trees growing under controlled environmental conditions before, during, and after drought and heat treatments. As observed in numerous other studies (e.g. Seco et al., 2015), net photosynthesis and isoprene emissions were coupled during non-stress conditions but became strongly uncoupled during heat and drought stress with substantial decreases in net photosynthesis but a stimulation of isoprene emissions.

General Comments The paper generally lacks any new biochemical and physiological mechanistic description of how isoprene and net photosynthesis can become uncoupled at high temperatures and drought. Thus, it is not clear what new information the new study adds other than reporting these expected results in a new tree species.

> **Reply:**
>
> We can understand the reviewer's concern regarding the comment on the novelty of our study, because we apparently did not highlight it well enough in the current version of the manuscript. We will do that in a revised version of the manuscript. So far, there is only one study (Vanzo et al., 2015) which evaluates isoprene emissions in response to prolonged combined and repeated heat-drought stress, we are thus addressing a poorly explored research area – despite of combined heat and drought being a feature of typical extreme episodic weather events which are likely to increase in future. Our study goes beyond usual leaf level measurements in that entire trees are exposed to elevated temperatures instead of controlling single leaves and in that ambient temperature variations were used to derive temperature response curves instead of switching between concrete temperature levels. In this manuscript we evaluated the stress-response of leaf-level emissions of four-year old black locust saplings and evaluated the change of temperature and light response functions of isoprene emissions, in view of alerting the modelling community to the complexity of the response patterns. To further strengthen this point and highlight the novelty of our study, we plan on adding two additional figures to a revised version of the manuscript (see Fig S1 and S2 at the end of the document).

However, some novel aspects of the work include a characterization of the light and temperature responses of isoprene emissions during stress. However, the very low light saturation of isoprene emissions of both control and stressed trees (200-300 micromol/m2/s) indicates that the plants were not adapted to normal high light conditions of plants in natural ecosystems during the growing season). As only leaves from the lower canopy were measured, it is difficult to understand how these results can be used for modeling of natural isoprene emissions from nature. Studies show that the majority of

photosynthesis and isoprene emissions from natural ecosystems occur in the upper canopy leaves exposed to full sunlight.

> **Reply:**
>
> We can understand the referee's concern but lower light levels under controlled compared to field conditions are a common phenomenon. However, there is no reason to think that this should have affected the different temperature responses of isoprene emissions as found in control versus stressed trees.
> The lower canopy measurements owe to the fast growth of black locust trees. Thus, the branch chambers which were initially installed in the mid to upper canopy (about 1.5 in height excluding pots) turned into lower canopy after some weeks of vigorous growth (trees were up to 5 m in height). Moreover, during prolonged stress preferentially the top-canopy leaves were shed, which would contradict top-of-canopy measurements.

Specific Comments: Take care when refereeing to photosynthesis; the measurements are of net photosynthesis not of gross rates of photosynthesis, which can be drastically different under high temperatures.

> **Reply:**
> We are aware of the difference between net and gross photosynthesis. The reviewer is correct that one needs to be specific in language used and clarify that we always refer to the net photosynthesis rate.

PTR-MS signals at m/z 69 are not necessarily unique to isoprene, especially under drought or high temperature where C5 green leaf volatiles can significantly contribute to their signal (Fall et al. 2001). Since GC measurements were not performed, the results cannot be considered quantitative.

> **Reply:**
> We disagree. Fall et al. (2001) showed that during drying of previously wounded leaves of non-isoprene emitters, C5 green leaf volatiles were identified at m/z ratio 69. Such compounds can add to the isoprene signal under certain circumstances which we can exclude in our study. We did not detect any leaf wounding, and also did not see an increase in acetaldehyde emissions which would be expected in case leaf wounding occurred. Moreover, in a similar study, Vanzo et al. (2015) reasoned that artificial cutting of leaves and the subsequent fast dehydration (for example after cutting grass) are not comparable to natural drought progression. In addition, we found isoprene emissions to quickly recover to the control treatment after stress release, which would not be the case if leaves were substantially harmed. For clarity to the reader we will add reference to these studies in a revised version of the manuscript.

Suggested Citations
Seco, R., Karl, T., Guenther, A., Hosman, K. P., Pallardy, S. G., Gu, L., Geron, C., Harley, P. and Kim, S. (2015), Ecosystem-scale volatile organic compound fluxes during an extreme drought in a broadleaf temperate forest of the Missouri Ozarks (central USA). Glob Change Biol, 21: 3657–3674. doi:10.1111/gcb.12980

> **Reply:**
> We will add the indicated reference to the revised manuscript.

**References:**

Fall, R., Karl, T., Jordan, A. and Lindinger, W.: Biogenic C5 VOCs: release from leaves after freeze–thaw wounding and occurrence in air at a high mountain observatory, Atmos. Environ., 35, 3905–3916, doi:10.1016/S1352-2310(01)00141-8, 2001.

Vanzo, E., Jud, W., Li, Z., Albert, A., Domagalska, M. A., Ghirardo, A., Niederbacher, B., Frenzel, J., Beemster, G. T. S., Asard, H., Rennenberg, H., Sharkey, T. D., Hansel, A. and Schnitzler, J.: Facing the Future: Effects of Short-Term Climate Extremes on Isoprene-Emitting and Nonemitting Poplar, Plant Physiol., 169, 560–575, doi:10.1104/pp.15.00871, 2015.

[Figure]

**Figure S1:** a) Isoprene fluxes of heat and heat-drought stressed trees modeled with the stress algorithm against fluxes modeled with the control algorithm including a linear least-square fit showing the slope which would bring fluxes calculated with the control algorithm in line with fluxes calculated with the stress algorithm; b) Isoprene fluxes modeled with the control algorithm and corrected with the slope denoted in S1a to account for changes in the standard isoprene emission rate during stress.

[Figure]

**Figure S2:** Modelled versus measured isoprene fluxes for trees exposed to control conditions (black circles), heat stress (red triangles), and heat-drought stress (blue squares) including a linear least square fit. Open grey symbols show isoprene fluxes modeled with the control algorithm instead of the corresponding algorithm for heat and heat-drought stressed trees.

---

## Author Comment (AC2) · 28 Apr 2017

**General comments:**

This manuscript makes the point that that isoprene emission model parameters are likely to be different for stressed plants than for unstressed plants. This is a good point to make.

> **Reply:**
> Thank you.

However, I found the manuscript to be problematic. Temperature of the individual leaves could not be controlled and so the temperature response curves of the control and heat or heat-drought treatments were almost non-overlapping.

> **Reply:**
> We understand the reviewer's concern which reflects a component of our study design: It was not our intent to directly control the temperature of individual leaves as the study was meant to mimic emission differences when the entire tree is exposed to a different temperature. Because we were interested in how trees responded to heat wave scenarios, we purposely mimicked diurnal temperature cycles and day-to-day variations. Since temperature responses are known to critically depend on how they were achieved (Niinemets et al., 2010) we intentionally choose this experimental design to mimic conditions how they could potentially occur during heat wave scenarios under ambient conditions (Boeck et al., 2010). The heat wave scenario was implemented on ambient temperatures with + 10°C on average – which, as the reviewer is correct, led to very different temperature range on the measured leaves. The reviewer is correct that this will need to be better highlighted and we will ensure that this becomes much clearer in a revised manuscript.

I found the description of the methods to be difficult. It is not clear to me whether leaves not currently being measured had an air flow of if the airflow only occurred during a measurement.

> **Reply:**
> Thank you for pointing us to this shortcoming of our methods description. The chambers (n=9 + 1 empty chamber), each permanently installed at one leaf petiole (see Fig. S3), were kept open all the time expect during the 10 min measurement, before which the chamber lids automatically closed. To ensure well mixing, the fan inside the chamber remained on at all times. Air flow (VOC-free) through the chamber, however, was only generated during measurements, while during the remaining time ambient air was mixed into the chamber. The permanent installation of the chambers enabled automation and excluded the risk of leaf wounding. We will make this much clearer in a revised version of the manuscript.

A great deal of variation in isoprene emission rates was observed. I was not convinced that the statistical treatments accurately reflected the variability. Isoprene emission is exceedingly difficult to predict, a point made by this lab that affects how these data need to be interpreted.

**Reply:**

As the reviewer states there is high variability in isoprene emission. This is due to changes in environmental drivers and tree-to-tree variability. However, besides this variability, differences between treatments were statistically significant as seen using linear-mixed effect models and uncertainty bounds for temperature response curves. Does the referee's comment further refer to the uncertainty bounds of the temperature and light fit, which might seem to be too low when compared to the variability of single measurement points? If this was the concern: as commonly done (see Seco et al., 2015) we used bin averaged data (each point weighted by the inverse of its standard deviation) to determine the fit for calculating the temperature and light response curves. The resulting confidence intervals of the fit (as shown in Figure 5) reflect that the fitted curves are statistically different above 25°C for heat trees and above 30°C for heat-drought trees (due to the higher variability in the heat-drought data). However, to clearly show that also the bin averaged isoprene emissions $E_{iso}$ are significantly different between stress and control treatments for T > 28°C we provide Table S1.

While a lot of work has gone into this report, I have significant concerns including that leaf temperature is not known and measurement temperature during isoprene emission measurement was almost non-overlapping.

**Reply:**

We can understand the reviewer's concern about the missing leaf temperature measurements. We purposely decided against controlling leaf temperature. Instead of leaf temperature we measured air temperature within the leaf cuvette because we were worried that attaching a thermocouple to a black locust leaf will provide us with data that might be difficult to interpret. This is (a) because black locust is known for leaf movement (paraheliotropism), during night-time leaves are generally folded and during daytime leaves fold when heat stress becomes severe (a hairy underside is exposed which helps to protect the leaves from excessive heating), and (b) which part of the pinnate leaf would be suited in order to measure average leaf temperature values? Thus, we could not be certain that a thermocouple attached to the underside would stay in place and deliver representable leaf temperature data. Air temperature is often used for the Guenther equations simply because leaf temperature is much harder to assess and often not at hand (Seco et al., 2015; Vanzo et al., 2015).

However, in order to add further information on this aspect we analyzed data of additional leaf temperature measurements (measured with an infrared camera, PI450 Optris GmbH, Berlin, Germany) during the second heat wave and compared air temperature to leaf temperature. Independent of treatment and temperature range, differences between air temperature and leaf temperature were found to be small and statistically not significant (see Table S2). Thus we are confident that in our study, air temperature is a good proxy for leaf temperature and plan to present Table S2 within the supplementary of a revised version of the manuscript.

A remark regarding the non-overlapping temperatures: The temperature curves of the control and heat-treated trees do not overlap because the control trees were not exposed to heat waves. We make sure that this becomes much clearer in the revised version of the manuscript.

The authors have an important point to make but the manuscript as written will not make that point very strongly. I made a number of comments on the pdf that I hope will be helpful to the authors.

**Reply:**
Thank you. We will modify a revised version of the manuscript to better highlight the differences in the temperature-light response equations as found for stressed black locust trees. We plan on including additional Figures (S1 and S1, see below) that clearly show that isoprene emissions during the heat waves would be overestimated (>50%) when parametrized with the control model. These figures together with an improved discussion would make our point much stronger in a revised version of the manuscript.

**Detailed Comments:**

Throughout the manuscript:

**Reply:**
All grammatical and style changes will be implemented in a revised version of the manuscript.

Line 151: Does this mean that the air flow through the chamber was turned off when it wasn't being measured?

**Reply:**
When the chamber was not measured it was open and thus circulated with ambient air (see detailed comment above and Fig. S3). This will be explained in more detail in a revised version of the manuscript.

Line 275: Is this the same as Es?

**Reply:**
No. We explicitly introduced a parameter EF (emission factor) for the temperature fit. In the original parametrization of the temperature response function for isoprene standard temperature was set to 27.8°C. Es (the standardized emission factor at 30°C) was thus not explicitly fitted in our case as we used the Guenther et al., (1991, 1993) over the Guenther (1997) equations as one parameter less was required for the fit. We will add Es to Table 3 in a revised version of the manuscript to improve comparability between literature results.

Line 296: How was leaf water potential measured?

**Reply:**
Mid-day leaf water potential was measured by determining the pressure necessary to cause water to exude from a freshly-cut leaf inserted in a Scholander pressure chamber (Model 1000, PMS Instrument Company, Albany, Oregon, USA,). We will add this information to a revised version of the manuscript.

Line 300: This sounds as though the heat-drought trees were subject to a different experiment the previous year than were the other trees.

**Reply:**
Thank you for pointing us to this. We re-word this sentence to make clear that the trees of the heat and heat-drought treatment had been subjected to two heat waves in the previous year, which resulted in a larger decline of basal area in the heat-drought (-43%) than heat-treated trees (-27%) (Ruehr et al., 2016). This reduction in biomass (reduced basal area and lower leaf biomass) may have

caused the relatively small differences in relative soil water content between the heat and the heat-drought treatment.

Line 317: It is hard to see this in 3c. The overwhelming impression is the variability.

**Reply:**
In this Figure, each data point reflects one measurement made with the automated leaf chamber, thus the variability includes diurnal variations and differences between the individual trees measured. The trees were exposed to ambient temperature changes and light fluctuations causing the pronounced variability in isoprene emissions the reviewer is referring to. We purposely decided to plot each single data point to highlight the difficulty of modeling isoprene emissions. However, alongside with the single data points we are also providing daily treatment averages, which clearly show the large increases in isoprene emissions. The significance of the treatment effects have also been confirmed by a linear mixed-effects model as given in Table 1 of the manuscript.
In a revised version of the manuscript we could omit the single data points and show instead the SE of the daily treatment averages - if this is preferred.

Line 355: Isoprene is very temperature dependent while photosynthesis is not but both a very light dependent. By restricting the analysis to less than 30oC the data would have been mostly in the light liming range for photosynthesis and so also for isoprene. Above 30°C light was probably mostly limiting and so the very different temperature responses of isoprene emission and photosynthesis would become dominant.

**Reply:**
We are not sure if we fully understood the reviewer comment. We assume that he/she refers to the differences in temperature responses of isoprene and photosynthesis which become more dominant for higher temperatures. Both, isoprene and photosynthesis are light dependent and temperature dependent. According to Ruehr et al., (2016) black locust trees in the control treatment were close to the temperature optimum (which was between 20°C and 30°C). Beyond this temperature optimum photosynthesis decreases with increasing temperatures. Since the amount of photosynthetic active radiation, to which trees were exposed, remained unchanged for heat-treated trees we conclude that the different temperature responses of photosynthesis and isoprene become more dominant for higher temperatures as photosynthesis already decreases while isoprene is still increasing. In a revised version of the manuscript this will be clarified in the Discussion.

Line 348: It isn't clear why this should be normalized to 500 when Es is normalized to 1000.

**Reply:**
Correct. In most studies Es is parameterized for light-saturation at 1000 µmol m$^{-2}$s$^{-1}$, however, the value used for standardization is an arbitrary value. In principle it does not matter to which light conditions Es is normalized as long as this value is above the light saturation for isoprene emissions. As in our study the photosynthetically active radiation hardly exceeded 500 µmol m$^{-2}$s$^{-1}$ and isoprene emissions reached its light saturation at values lower than 500 µmol m$^{-2}$s$^{-1}$ we used this value for normalization. We will explain our considerations in a revised version of the manuscript.

Line 353: I am not good at statistics, but I don't feel these numbers represent what I would intuitively take from Fig 5. The highest rate of the bin averaged control seems to exceed any control measurements. The table confirms that there was only one measurement above 32°C. The extreme variability, especially of the heat-drought treatment, make it difficult to draw specific conclusions. I would only conclude that very high rates are possible in heat-drought but low rates are also possible, possibly reflecting dying leaves.

> **Reply:**
> The bin-averages 32–35°C and 35-40°C in the control treatment consist of one measurement point each. Since these points did pass the quality control, we did not exclude them from the fit, but used a lower weight which was calculated for each bin average using the inverse standard deviation (in case of n=1, SD was artificially set to 100; we will add this to the Methods section in a revised manuscript)
> In this case SE refers to the fit of EF, since the uncertainty of this parameter in the fit was relatively low (at least for the control and heat data) we got a relatively low standard error. Please consider that we did fit the curves to bin averaged data points and not to single measurement points (which are shown for reasons of transparency). We prepared Table S1 with bin averages and corresponding standard errors to show that bin averaged isoprene emissions for T > 28°C are significantly different (based on a t-test) between the control and the stress treatments as well. In case it is required, we can include this table also into the supplementary of a revised version of the manuscript.

Line 354: %a is a temperature response and not normalized data

> **Reply:**
> We apologize for this mistake: we refer to Fig 5b here and will change that in a revised version of the manuscript.

Line 375: Both isoprene synthase and DMAPP availability affect this as recent papers have shown.

> **Reply:**
> Thank you for pointing us to this. We will change that accordingly in a revised version of the manuscript.

Line 386: First seen by LoretoSharkey TD, Loreto F (1993) Water stress, temperature, and light effects on the capacity for isoprene emission and photosynthesis of kudzu leaves. Oecologia95, 328-333.

> **Reply:**
> We will add this reference in a revised version of the manuscript.

Line 397: I would also cite the work of Delwiche Delwiche CF, Sharkey TD (1993) Rapid appearance of 13C in biogenic isoprene when $^{13}CO_2$ is fed to intact leaves. Plant, Cell & Environment16, 587-591

> **Reply:**
> We will add this reference in a revised version of the manuscript.

Line 404: Sharkey and Loreto saw 67% Sharkey TD, Loreto F (1993) Water stress, temperature, and light effects on the capacity for isoprene emission and photosynthesis of kudzu leaves. Oecologia 95, 328-333.

> **Reply:**
> Thank you. In a revised version of the manuscript we will change that accordingly.

Line 436: I am not convinced of this

> **Reply:**
> We have taken this information from the 95% confidence intervals of the fit derived for the control and the heat and heat-drought treatment (these do not overlap completely). In a revised version of the manuscript we will add Fig S1 and Fig S2 which illustrate differences between measured data and the treatments fitted curves in in a better way. Thus, we will change this part of the discussion accordingly to make our point much clearer.

Line 461: This is new and likely to well accepted but the current manuscript does not make a strong enough case for it.

> **Reply:**
> Thank you. As mentioned before, we will make this much stronger in a revised version of the manuscript by including new Figures (Fig S1 and S2) and a revised Results and Discussion section accordingly.

**References:**

Boeck, H. J. De, Dreesen, F. E., Janssens, I. A. and Nijs, I.: Climatic characteristics of heat waves and their simulation in plant experiments, Glob. Chang. Biol., 16, 1992–2000, doi:10.1111/j.1365-2486.2009.02049.x, 2010.

Niinemets, Ü., Monson, R. K., Arneth, a., Ciccioli, P., Kesselmeier, J., Kuhn, U., Noe, S. M., Peñuelas, J. and Staudt, M.: The leaf-level emission factor of volatile isoprenoids: caveats, model algorithms, response shapes and scaling, Biogeosciences, 7(6), 1809–1832, doi:10.5194/bg-7-1809-2010, 2010.

Seco, R., Karl, T., Guenther, A., Hosman, K. P., Pallardy, S. G., Gu, L., Geron, C., Harley, P. and Kim, S.: Ecosystem-scale volatile organic compound fluxes during an extreme drought in a broadleaf temperate forest of the Missouri Ozarks (central USA), Glob. Chang. Biol., 21, 3657–3674, doi:10.1111/gcb.12980, 2015.

Vanzo, E., Jud, W., Li, Z., Albert, A., Domagalska, M. A., Ghirardo, A., Niederbacher, B., Frenzel, J., Beemster, G. T. S., Asard, H., Rennenberg, H., Sharkey, T. D., Hansel, A. and Schnitzler, J.: Facing the Future: Effects of Short-Term Climate Extremes on Isoprene-Emitting and Nonemitting Poplar, Plant Physiol., 169, 560–575, doi:10.1104/pp.15.00871, 2015.

**Table S1**: Bin averaged isoprene emissions ($E_{iso}$) for different temperature classes including the corresponding standard errors (if n>1). Average values highlighted with **\*** denote that $E_{iso}$ is significantly different ($p<<0.05$ based on a t-test if the number of measurements exceeded three) to the average in the control group ($^{(*)}$ denotes that $E_{iso}$ between both stress treatments are significantly different).

| Treatment | Control | | Heat | | Heat-drought | |
|---|---|---|---|---|---|---|
| Temperature range (°C) | $E_{iso}$ (nmol m$^{-2}$s$^{-1}$) | SE | $E_{iso}$ (nmol m$^{-2}$s$^{-1}$) | SE | $E_{iso}$ (nmol m$^{-2}$s$^{-1}$) | SE |
| 15-20 | 1.52 | 0.01 | - | - | - | - |
| 20-24 | 3.81 | 0.42 | - | - | - | - |
| 24-28 | 7.33 | 0.76 | 4.95 | 1.23 | 9.83 | 2.31 |
| 28-32 | **16.14** | **1.81** | **9.13\*** | **0.81** | **10.50\*** | **1.17** |
| 32-35 | 23.12 | - | 16.62 | 0.82 | 18.83 | 1.82 |
| 35-40 | 40.41 | - | 26.26 | 1.03 | 24.86 | 1.83 |
| 40-45 | - | - | **37.43$^{(*)}$** | **2.58** | **23.36$^{(*)}$** | **2.60** |
| >45 | - | - | 32.35 | - | 13.54 | 4.24 |

**Table S2**: Air temperature and corresponding leaf temperature including standard errors measured for n tree leafs in the control, heat, and heat-drought treatment. Leaf temperature was measured with an infrared camera on two days during the second heat wave. Differences between leaf and air temperature were not significant (pairwise t-test p >0.05).

| Treatment | $T_{air} \pm SE$ (°C) | $T_{leaf} \pm SE$ (°C) | $T_{leaf} - T_{air}$ (°C) | $p < 0.05$ | n |
|---|---|---|---|---|---|
| **Control** | 21.8 ± 1.9 | 21.1 ± 1.9 | 0.7 ± 0.5 | n.s. | 3 |
| **Heat** | 34.7 ± 1.2 | 34.4 ± 1.5 | -0.3 ± 0.5 | n.s. | 5 |
| **Heat-drought** | 36.0 ± 0.5 | 35.2 ± 0.8 | -0.9 ± 0.5 | n.s. | 10 |

[Figure]

**Figure S1:** a) Isoprene fluxes of heat and heat-drought stressed trees modeled with the stress algorithm against fluxes modeled with the control algorithm including a linear least-square fit showing the slope which would bring fluxes calculated with the control algorithm in line with fluxes calculated with the stress algorithm; b) Isoprene fluxes modeled with the control algorithm and corrected with the slope denoted in S1a to account for changes in the standard isoprene emission rate during stress.

[Figure]

**Figure S2:** Modelled versus measured isoprene fluxes for trees exposed to control conditions (black circles), heat stress (red triangles), and heat-drought stress (blue squares) including a linear least square fit. Open grey symbols show isoprene fluxes modeled with the control algorithm instead of the corresponding algorithm for heat and heat-drought stressed trees.

[Figure]

**Fig S3:** Picture of a leaf chamber enclosing a black locust leaf. The lid closed automatically during measurements for about 10 minutes. Between measurements the lid remained open and a fan was circulating air constantly.

---

## Author Comment (AC3) · 28 Apr 2017

Summary:

This work examines the changes in foliar isoprene emission and photosynthesis of isoprene emitting black locust to periods of drought and drought plus heat stress. The isoprene and photosynthesis responses are compared to existing literature and the response of isoprene emission was then compared to that which would be calculated using the Guenther et al algorithm under the same stress conditions. In general the study is well written, although some aspects could be clearer (outlined in the comments sections below). There are a few typographical errors and the within text references need to be looked at as they should be presented either alphabetically or by date order. This study does appear to contain a solid body of work which is worth publishing to add to our understanding of isoprene responses to complex stresses and to help improve modelled emission estimates. However, as the manuscript is currently written I struggle to find the novelty in the work. I have a few concerns and believe the manuscript should be improved as outlined below before it could be accepted for publication.

> **Reply:**
> We like to thank the reviewer for considering the quality of our study. Regarding the comment on the novelty of our study, we can understand the reviewer's concern because we apparently did not highlight it well enough in the current version of the manuscript.
>
> We will do that in a revised version of the manuscript. So far, there is only one study (Vanzo et al., 2015) which evaluates isoprene emissions in response to prolonged combined and repeated heat-drought stress, we are thus addressing a poorly explored research area – despite of combined heat and drought being a feature of typical extreme episodic weather events which are likely to increase in future. In this manuscript we evaluated the stress-response of leaf-level emissions of four-year old black locust saplings and evaluated the change of temperature and light response functions of isoprene emissions, in view of alerting the modelling community to the complexity of the response patterns. See also our reply to major comment # 4, below, for more detail.

Major comments:

1) Materials and Methods, Experimental set up, line 96. I am concerned that the trees in the stress treatments had previously been exposed to two experimental heat waves and were showing a difference in basal area. Previous work has shown that VOC emissions differ based on exposure to previous environmental conditions (e.g. Sharkey et al, 1999 and citing references). Could the authors provide some reassurance that after pruning and over wintering the development and growth rates were then equivalent and could be fairly compared to one another? If they were not equivalent as suggested in the results section 3.1, were the data normalised?

**Reply:**

The data were collected as part of a full three-years experiment which sought to evaluate the response to prolonged and repeated stress. During the first year it was unfortunately not possible to collect VOC data, but information from the first year of the experiment showed that black locust leaves recovered its photosynthesis 3 weeks after the last heat wave ended and that basal growth rates were close to control trees (Ruehr et al. 2016). Trees were pruned due to height constraints in greenhouse facility and overwintered outside. Before leaf-out in spring, the trees were returned inside the greenhouse and equipped with sensors. Branch chambers were installed in June. Statistical analysis showed no differences in leaf gas exchange (photosynthesis and isoprene emission) before the heat-waves were imposed in the second year of the experiment (see Table 2 of the current manuscript). Therefore we do not think it would be necessary to normalize the data and we are confident that leaf level emissions did not carry a substantial signal as a consequence of the stress during the first year of the experiment. We will include these aspects to a revised version of the manuscript.

2) Could the authors give an explanation as to why the trees were not randomly selected for the work included in the current study? This would have given a mixture of previously stressed and unstressed trees in each treatment group and removed any concern that the prior treatment of these trees was affecting the current results.

**Reply:**

This is an important point, and reflects the study design over the entire duration. The purpose was to evaluate how the trees will response to re-occurring heat waves over subsequent years – which made it necessary to maintain trees within one treatment. Studies on heat waves occurring over more than one growing season are scarce and to our knowledge have not been done yet with woody species. Although we found a slightly reduced basal area of previously heat and heat-drought stressed trees in the second year of the experiment, we detected no change in leaf-level emissions of newly grown leaves of stressed trees compared to the control prior to the second year heat waves' (see LME results in Table 2 of the manuscript).

3) I would also like to see a clear description of the growth conditions and number of trees used per treatment and per measurement. Could the authors give a full description of the growth conditions of the trees (temperature, light, CO2, RH) in the description of the experimental setup? Did the greenhouse have supplemental lighting, where was average PAR recorded, what was the day length? How many replicates were used per measurement? At the moment it is not clear to me how many replicates were used for what.

**Reply:**

The reviewer is correct in that we did not provide all this information in the Methods section, but instead referred the reader to a publication that describes the experimental set-up in great detail. In order to facilitate reading of the manuscript, we plan to add more detail on the methods into a revised manuscript:

In total we had six trees per treatment, however, leaf chambers were installed at three trees per treatment. Although the major component of the photosynthetic active radiation was the sunlight, the greenhouse had supplemental lighting (Philips SON-T Agro 400 W, Philips, Amsterdam, NL). Daylight-length was not artificially modulated, and thus varied according to season. Growth conditions were monitored by two sensors per greenhouse compartment measuring photosynthetic active radiation (PQS 1, Kipp & Zonen, Delft, The Netherlands), air temperature, and relative humidity (CS215, Campbell Scientific Inc., Logan, UT, USA). Photosynthetically active radiation as used for the light response curves was recorded alongside temperature in each leaf chamber. Average growth conditions (PAR, VPD, and temperatures) for the trees within the two compartments of the greenhouse are presented in Duarte et al. (2016). However, for clarity we plan to include a table presenting average $CO_2$ concentrations, temperatures, relative humidity, and photosynthetic active radiation (Table S3) monitored in the greenhouse compartments in a revised version of the manuscript. In general none of the drivers differed by more than 2 % between the two greenhouse compartments before the heat waves.

4) In general I cannot currently see the novelty of this work. However, this might be improved if the authors could use their data to suggest a new algorithm or an amendment to the existing algorithm to bring modelled isoprene emissions more in line with that which is observed. At the moment the authors highlight the difference between the observed and modelled emissions but don't go any further.

**Reply:**
Regarding the novelty of our study, please see also our answer above and comments to reviewer 1 and 2. More specifically, to highlight the complexity of modelling isoprene emissions under combined stress and recovery with simple algorithms we prepared two additional figures (Fig S1 and S2) showing much clearer the differences between the stress and control model in estimating isoprene emission in black locust. When using the model parameterized based on data from the control trees, heat and heat-drought isoprene emissions would be overestimated by approximately 50 %. While past environmental conditions are known to alter the isoprene emission factor (Niinemets et al., 2010), we found indication that stressful conditions will as well alter the shape of the temperature response function (e.g. temperature maximum of the response curves moves towards a higher temperature). Thus it is not possible to apply a simple correction factor (based on the slope in Fig S1a) to adjust the standardized emission rate (compare Fig S1b) and bring measured emissions in line with emissions modeled with the control model including a correction factor.
This together with our reasoning before will be added to a revised version of the manuscript.

Minor comments
1) Abstract line 12 – mentions assessing the impact of stress on BVOC emissions but only isoprene is presented in the manuscript. Either remove the reference to general BVOC or include other emitted compounds.

**Reply:**

Thank you. We will change the wording accordingly

2) Intro, line 38 – include ref to more recent Wyche et al, ACP 2014 which gives positive and negative effects of isoprene emission on secondary aerosol formation.

**Reply:**

Thank you for pointing us to this reference. We will include it in a revised version of the manuscript

3) Into, line 65 and line 71 – include ref to more recent Ryan et al, New Phyt 2014 and remove older references unless they are seminal /original work.

**Reply:**

We will include the more recent literature and remove older literature which is not original work in a revised version of the manuscript.

4) Mat and Methods, Paragraph starting line 155 – description is not clear. Is the automatic switching of the measurements or the air flow? If air flow does this mean the chambers were clamped on the plants with no air flow for a period of time?

**Reply:**

Thank you for pointing us to this shortcoming of our methods description. The chambers (n = 9 +1 empty chamber), each permanently installed at one leaf petiole (see Fig S3), were kept open all the time expect during the 10 min measurement, before which the chamber lids automatically closed. To ensure well mixing, the fan inside the chamber remained on at all times. Air flow (VOC-free) through the chamber, however, was only generated during measurements, while during the remaining time ambient air was mixed into the chamber. The permanent installation of the chambers enabled automation and excluded the risk of leaf wounding. We will make this much clearer in a revised version of the manuscript.

5) Section VOC Line 200 – the PTR-MS only counts set masses and cannot give compound identification. Could the authors include information on any mass identification that was performed (e.g. GC-MS) to confirm that it was only isoprene at m/z 69

**Reply:**

It is true that the PTR-MS only counts nominal masses. Since black locust is known to be a relatively strong isoprene emitter we are confident that in our case, as well as in other studies (see Vanzo et al., 2015) the signal on m/z 69 is due to isoprene. Please also see our answer to reviewer 1. We will explain that in more detail in a revised version of the manuscript.

6) Line 231 – 500 PAR seems quite low for trees in the summer. Top of canopy PAR in

northern Europe during the summer is more likely to be between 1000 and 2000 PAR. Could the authors give a reason for choosing 500 PAR.

**Reply:**
Correct. In most studies Es is parameterized for light-saturation at 1000 µmol m$^{-2}$s$^{-1}$, however, the value used for standardization is an arbitrary value. In principle it does not matter to which light conditions Es is normalized as long as this value is above the light saturation for isoprene emissions. As in our study the photosynthetic active radiation hardly exceeded 500 µmol m$^{-2}$s$^{-1}$ and isoprene emissions reached its light saturation at values lower than 500 µmol m$^{-2}$s$^{-1}$ we used this value for normalization. We will explain our considerations in a revised version of the manuscript.

7) Mat & Methods Line 267 - Formatting error

**Reply:**
This will be corrected in a revised version of the manuscript.

8) Results 3.1 line 295. Could the authors include a description of how midday leaf water potential was measured?

**Reply:**
Mid-day leaf water potential was measured by determining the pressure necessary to cause water to exude from a freshly-cut leaf inserted in a Scholander pressure chamber (Model 1000, PMS Instrument Company, Albany, Oregon, USA,). We will add this information to a revised version of the manuscript.

9) Results 3.1 line 299 – typo "relative" should be "relatively"

**Reply:**
Thank you for catching this. The typo will be corrected in a revised version of the manuscript.

10) Results 3.2 line 307 – I don't understand why " (PAR > 50 umol m-2 s-1)" is included in this sentence, when the sentence is referencing stomatal conductance – please clarify.

**Reply:**
To clarify that we explicitly calculated daytime averages, since stomatal conductance is nearly zero during the night the averaging period makes a difference in the results. This will be explained in a revised version of the manuscript.

11) Line 316 Daytime (PAR > 50 umol m-2 s-1) – I am assuming this means the authors collated any data collected when PAR readings were over this value to be "daytime"

values. If this is correct please include a clarification at first use to make it easier for the reader to understand.

**Reply:**
We will add a sentence to the revised version of the manuscript to make this clear.

12) Line 322 – It may be over-stretching the results to include "marginally significant (p value around 0.1)" results as significant differences. This is not common practice but is perhaps personal preference.

**Reply:**
We wanted to indicate that the p-value suggests that these values tend to be higher compared to the control even if the change is not significant based on the $p < 0.05$ criterion. We change our wording accordingly.

13) Results 3.3, line 338 "significantly different to control trees" and "no significant differences. . ." please give p values.

**Reply:**
Agreed. We will add the corresponding p-values in a revised version of the manuscript. The decision was based on the criteria $p < 0.05$ (which is common practice).

14) Discussion Line 380 – references you should include more recent ref e.g. Ryan et al New Phyt 2014 who used genetically modified tobacco specifically to study the impact of drought on isoprene emission and protection.

**Reply:**
Thank you. We will make sure to include more recent literature in a revised version of the manuscript.

15) Line 385 "A quick recovery of isoprene emissions after periods of drought stress seems to emerge as a 385 common feature that has also been observed in previous studies (Brilli et al., 2013; Pegoraro et al., 2004; Velikova and Loreto, 2005)" and line 288 "The observed faster recovery of isoprene emissions than photosynthesis may be a common pattern following stress release (Brilli et al., 2013; Pegoraro et al., 2004)." This appears to be a repeated point – please remove one of the sentences.

**Reply:**
We will critically re-assess our wording and make sure that the intended differences between the sentences become clear.

16) Line 390 – "this is the first study that considers dynamics of isoprene emissions during and following combined heat–drought stress. . ." Unfortunately this claim is untrue – please remove and see Vanzo et al, 2015 and references therein.

**Reply:**

We apologize for this mistake and will change the sentence accordingly following our reasoning given above and of course include this reference in a revised version of the manuscript.

17) Paragraph beginning line 415 – including reference to Ryan et al, 2014, New Phytologist, who studied isoprene emitting and non-emitting plant responses to drought, would be appropriate here. Most likely with the Vickers et al, 2009 reference. 18) Table 2–could the authors explain why there is such a variation in group sizes (n values from 0–49)?

**Reply:**

This has two reasons:

(1) As we did not randomize temperatures in the heat and heat-drought treatment but simulated high temperatures using ambient +10°C we do not have an equal number of points in all temperature levels. Thus in some temperature regimes there is a dense distribution of points while in others there are less points.

(2) Since we wanted to use the same temperature bins for all three treatments (which makes the bin averages more comparable between treatments) it was not possible to set bins in such a way that we have in every temperature range a similar group size. Especially for the highest and lowest temperatures of each treatment we thus have a lower number of points within the bin. To account for this we weighted the bin averages by standard deviation. We will take care that this information will become clearer in the Methods section.

**References**

Duarte, A. G., Katata, G., Hoshika, Y. and Hossain, M.: Immediate and potential long-term effects of consecutive heat waves on the photosynthetic performance and water balance in Douglas-fir, J. Plant Physiol., 205, 57–66, doi:10.1016/j.jplph.2016.08.012, 2016.

Niinemets, Ü., Arneth, A., Kuhn, U., Monson, R. K., Penuelas, J. and Staudt, M.: The emission factor of volatile isoprenoids: stress, acclimation, and developmental responses, Biogeosciences, 7, 2203–2223, doi:10.5194/bg-7-2203-2010, 2010.

Ruehr, N. K., Gast, A., Weber, C., Daub, B. and Arneth, A.: Water availability as dominant control of heat stress responses in two contrasting tree species, Tree Physiol., 36(2), 164–178, doi:10.1093/treephys/tpv102, 2016.

Vanzo, E., Jud, W., Li, Z., Albert, A., Domagalska, M. A., Ghirardo, A., Niederbacher, B., Frenzel, J., Beemster, G. T. S., Asard, H., Rennenberg, H., Sharkey, T. D., Hansel, A. and Schnitzler, J.: Facing the Future: Effects of Short-Term Climate Extremes on Isoprene-Emitting and Nonemitting Poplar, Plant Physiol., 169, 560–575, doi:10.1104/pp.15.00871, 2015.

**Table S3**: Average $CO_2$ concentration, temperature, relative humidity (RH), and daytime photosynthetically active radiation (PAR>100 µmol $m^{-2}s^{-1}$) including the corresponding standard deviation in the two greenhouse compartments between 7 May and 13 June 2014, before the start of the first heat wave. Difference in growth conditions between the average values of environmental drivers are given in percent.

| growth conditions 07.05.14 – 13.06.14 | compartment 1 | | compartment 2 | | |
|---|---|---|---|---|---|
| | average | standard deviation | average | standard deviation | difference (%) |
| $CO_2$ (ppm) | 409 | 39 | 404 | 36 | 1.2 |
| Temperature (°C) | 15.6 | 5.4 | 15.6 | 5.2 | 0 |
| RH (%) | 80.8 | 13.8 | 82.1 | 12.8 | 1.6 |
| daytime PAR (µmol $m^{-2}s^{-1}$) | 419 | 286 | 412 | 248 | 1.9 |

[Figure]

**Figure S1:** a) Isoprene fluxes of heat and heat-drought stressed trees modeled with the stress algorithm against fluxes modeled with the control algorithm including a linear least-square fit showing the slope which would bring fluxes calculated with the control algorithm in line with fluxes calculated with the stress algorithm; b) Isoprene fluxes modeled with the control algorithm and corrected with the slope denoted in S1a to account for changes in the standard isoprene emission rate during stress.

[Figure]

**Figure S2:** Modelled versus measured isoprene fluxes for trees exposed to control conditions (black circles), heat stress (red triangles), and heat-drought stress (blue squares) including a linear least square fit. Open grey symbols show isoprene fluxes modeled with the control algorithm instead of the corresponding algorithm for heat and heat-drought stressed trees.

[Figure]

**Fig S3:** Picture of a leaf chamber enclosing a black locust leaf. The lid closed automatically during measurements for about 10 minutes. Between measurements the lid remained open and a fan was circulating air constantly.

---

## Author Response (AR1)

**Dear Editor,**

We would like to thank the editor and all the reviewers for their thoughtful comments and advices how to improve our manuscript "*Isoprene emission and photosynthesis during heat waves and drought in black locust*". We carefully revised the manuscript by addressing each of the editor's and reviewers' comments (find the detailed replies below). In particular, we added:

- two new Figures (Fig. 6 and 7) to the manuscript addressing the differences between model parameterization using standard conditions (control) versus treatment-specific models and provide information on changes in emission factors and the shape of temperature response functions under stress conditions. We modified the Abstract accordingly.
- more information to the experimental setup and motivation for study design, revised Fig. 1 and added a new Table 1 with the environmental conditions before stress.
- an appendix including a comparison of air temperature with leaf temperature and bin-averages of isoprene emissions for each treatment
- text to the introduction and discussion to highlight the novelty of the study

Changes to the manuscript are highlighted. Please find our detailed point-to-point answers to each of the comments below.
We are convinced that the revised manuscript improved considerably and hope it is now suitable for publication in Biogeosciences.

**Best regards,**

Ines Bamberger

**Editor Comments:**

Thank you for your thoughtful response to the three reviews we received on your manuscript. As you can see, each reviewer had quite useful comments on ways in which to improve your manuscript. In particular, I would echo the need to highlight the novelty our your results, and encourage you to not only draw attention to the Vanzo et al. paper but also discuss how your results support or add those presented there.

> **Reply:**
> We have rewritten substantial parts of our introduction to highlight the novelty of our study (line 76–86 and 94-97). We also have made it much clearer that our study differs from others (especially heat-stress studies) in that we have exposed the trees to close to natural temperature fluctuations by mimicking outside air temperature (conditions during heat waves were outside temperature +10°C). This study design was explicitly chosen to draw the attention of the modeling community to the complexity of isoprene emissions under closer-to-natural conditions.
>
> We put our study in context to Vanzo et al. (2015; see line 471-482) and discuss the potential of isoprene emissions protecting leaves during heat stress and their role in enabling a quick recovery after stress release.

Your proposed figures S1 and S2 are a great addition and should be very useful for doing so. Concerns were repeatedly raised about the insufficient detailing of the experimental set-up, so please endeavor to clearly state not only what was done but also what motivated your choices.

The fact that leaf temperature was not controlled for is potentially problematic, but I agree with your response and feel these concerns should be alleviated in part by a more detailed description of the experimental set-up.

**Reply:**
We added the two Figures S1 and S2 (Figure 6 and 7 in the revised manuscript) and extended results and discussion accordingly. We compare now with other studies, which also found that emission factors can change during stress. Additionally we found that not only $E_S$ changed, but that the shape of the temperature response functions (determined by $C_{T1}$, $C_{T2}$ and $T_m$) differed between control and stress treatments. In the revised manuscript we provide more information on these findings (results section line 424-425 and discussion line 549-572). Our study expands to recent findings on temperature-response parameters under drought (Geron et al. 2016) in that we report these changes during pronolonged heat and heat-drought stress. We highlight this in several instances in the manuscript.

Regarding the experimental set-up, we added more details to the description of the experimental setup (e.g. 122-133) and made clear why it was not possible to control leaf temperature in our study (line 189-196). As mentioned before, we explicitly wanted to address temperature variations as can be expected under closer-to-natural conditions (line 94-97), therefore the temperature control of the greenhouse was set to mimic ambient air temperature (measured in front of the greenhouse, see line 133-136); during heat waves +10°C were added. To alleviate concerns of differences in leaf and air temperature we added Table A1, which shows only small differences (< 1°C) on the two occasions measured during the second heat wave, when an infrared camera was at hand.

Finally, the difference in canopy position highlighted by reviewer 1 will certainly lead to differences in at least the basal rate of emissions, as leaf physiology is highly acclimated to light environment. This issue warrants some discussion in order to help the reader in their interpretation of your results.

**Reply:**
We added a sentence to the results section to explicitly point out that the response curves reached light saturation quite early as a consequence of the relatively low light levels in the greenhouse and that measurements may therefore be more comparable to mid or lower canopy conditions. '*Compared to literature values isoprene emissions of all trees reached light saturation at relatively low values of photosynthetic active radiation (e.g. PAR between 200 and 300 µmol m 2s 1 for the control and heat–drought stressed trees), most probably because of the relatively low levels of PAR (see Table 1) in the greenhouse. Response curves of isoprene versus environmental conditions obtained in this study will thus be more comparable to lower canopy conditions where leaves are not constantly under light saturation. To reflect this, we normalized temperature responses of isoprene to a PAR of 500 µmol m$^{-2}$ s$^{-1}$ (typically values of 1000 µmol m$^{-2}$ s$^{-1}$ are used in the literature).*' We hope this will help the reader to interpret the results.

**Common points of the reviewers:**

**Regarding the novelty:**

Referee #1: The paper generally lacks any new biochemical and physiological mechanistic description of how isoprene and net photosynthesis can become uncoupled at high temperatures and drought. Thus, it is not clear what new information the new study adds other than reporting

these expected results in a new tree species. However, some novel aspects of the work include a characterization of the light and temperature responses of isoprene emissions during stress.

Referee #2: The authors have an important point to make but the manuscript as written will not make that point very strongly. I made a number of comments on the pdf that I hope will be helpful to the authors.

Referee #3: This study does appear to contain a solid body of work which is worth publishing to add to our understanding of isoprene responses to complex stresses and to help improve modelled emission estimates. However, as the manuscript is currently written I struggle to find the novelty in the work.
In general I cannot currently see the novelty of this work. However, this might be improved if the authors could use their data to suggest a new algorithm or an amendment to the existing algorithm to bring modelled isoprene emissions more in line with that which is observed. At the moment the authors highlight the difference between the observed and modelled emissions but don't go any further.

> **Reply:**
>
> We can understand the reviewer's concern regarding the comment on the novelty of our study, because we apparently did not highlight it well enough in the current version of the manuscript.
> To highlight the novelty of our study we

(i)    Revised parts of the introduction that will now lead the reader more directly to the novel points of this study: Line 76–86 and 94-97

(ii)   We included two new figures (Fig. 6 and 7) and added:
> Section 3.5 'Modeled leaf-level isoprene emissions' to the results and
> Section 4.4 'Modelling isoprene emissions during stress' to the discussion.
> These changes points highlight the novel points of the study and improve the manuscript.

**Regarding the Method description:**

Referee #2 I found the description of the methods to be difficult. It is not clear to me whether leaves not currently being measured had an air flow of if the airflow only occurred during a measurement.

Referee #3 I would also like to see a clear description of the growth conditions and number of trees used per treatment and per measurement. Could the authors give a full description of the growth conditions of the trees (temperature, light, $CO_2$, RH) in the description of the experimental setup? Did the greenhouse have supplemental lighting, where was average PAR recorded, what was the day length? How many replicates were used per measurement? At the moment it is not clear to me how many replicates were used for what.

> **Reply:**
>
> In order to facilitate reading of the manuscript, we added more detail on the methods (Section 2.1, line 113-118 and line 122-136) into a revised manuscript, revised passages which were unclear (Section 2.1.1, line 166-172), and added Table 1 which illustrates the growth conditions in the two compartments of the greenhouse.

**Other points:**
**Anonymous Referee #1**

However, some novel aspects of the work include a characterization of the light and temperature responses of isoprene emissions during stress. However, the very low light saturation of isoprene emissions of both control and stressed trees (200-300 micromol/m2/s) indicates that the plants were not adapted to normal high light conditions of plants in natural ecosystems during the growing season). As only leaves from the lower canopy were measured, it is difficult to understand how these results can be used for modeling of natural isoprene emissions from nature. Studies show that the majority of photosynthesis and isoprene emissions from natural ecosystems occur in the upper canopy leaves exposed to full sunlight.

> **Reply:**
> We can understand the referee's concern but lower light levels under controlled compared to field conditions are a common phenomenon.
> The lower canopy measurements owe to the fast growth of black locust trees. Thus, the branch chambers which were initially installed in the mid to upper canopy (about 1.5 in height excluding pots) turned into lower canopy after some weeks of vigorous growth (trees were up to 5 m in height). Moreover, during prolonged stress preferentially the top-canopy leaves were shed, which would contradict top-of-canopy measurements.
> We added some sentences to the results (section 3.4.) which refer to the drawback of the relatively low photosynthetically active radiation (which is a result of measurements within the greenhouse) and explains that these measurements will be more comparable lower canopy conditions '*Compared to literature values isoprene emissions of all trees reached light saturation at relatively low values of photosynthetic active radiation (e.g. PAR between 200 and 300 $\mu mol\ m^{-2}s^{-1}$ for the control and heat–drought stressed trees), most probably because of the relatively low levels of PAR (see Table 1) in the greenhouse. Response curves of isoprene versus environmental conditions obtained in this study will thus be more comparable to lower canopy conditions where leaves are not constantly under light saturation. To reflect this, we normalized temperature responses of isoprene to a PAR of 500 $\mu mol\ m^{-2}s^{-1}$ (typically values of 1000 $\mu mol\ m^{-2}s^{-1}$ are used in the literature).*'

Specific Comments: Take care when refereeing to photosynthesis; the measurements are of net photosynthesis not of gross rates of photosynthesis, which can be drastically different under high temperatures.

> **Reply:**
> We changed the wording to net photosynthesis throughout the manuscript.

PTR-MS signals at m/z 69 are not necessarily unique to isoprene, especially under drought or high temperature where C5 green leaf volatiles can significantly contribute to their signal (Fall et al. 2001). Since GC measurements were not performed, the results cannot be considered quantitative.

> **Reply:**
> For clarity to the reader we explained this possible interference of C5-compounds which could under certain circumstances contribute to the PTR-MS signal at m/z 69 and why we are convinced that in our study this is not the case (Section 2.1.3, line 247-254).

Suggested Citations
Seco, R., Karl, T., Guenther, A., Hosman, K. P., Pallardy, S. G., Gu, L., Geron, C.,
Harley, P. and Kim, S. (2015), Ecosystem-scale volatile organic compound fluxes during

an extreme drought in a broadleaf temperate forest of the Missouri Ozarks (central USA). Glob Change Biol, 21: 3657–3674. doi:10.1111/gcb.12980

> **Reply:**
> We added this reference to the revised version of the manuscript.

**Anonymous Referee #2**

**General comments:**

However, I found the manuscript to be problematic. Temperature of the individual leaves could not be controlled and so the temperature response curves of the control and heat or heat-drought treatments were almost non-overlapping.

> Reply:
> We understand the reviewer's concern which reflects a component of our study design: It was not our intent to directly control the temperature of individual leaves as the study was meant to mimic emission differences when the entire tree is exposed to a different temperature. Because we were interested in how trees responded to heat wave scenarios, we purposely mimicked diurnal temperature cycles and day-to-day variations. Since temperature responses are known to critically depend on how they were achieved (Niinemets et al., 2010) we intentionally choose this experimental design to mimic conditions how they could potentially occur during heat wave scenarios under ambient conditions (Boeck et al., 2010). The heat wave scenario was implemented on ambient temperatures with + 10°C on average – which, as the reviewer is correct, led to very different temperature range on the measured leaves.
> We made our intentions much clearer in the revised version of the manuscript and added reasoning to the introduction (line 76-86 and line 94-97). Additionally we stated why it was not possible to directly measure or even control leaf temperature (Section 2.1.1, last paragraph), and to alleviate concerns in differences between air and leaf temperature added information on supplementary measurements (Table A1). Hence, it should be clear now that we have chosen on purpose close-to-natural temperature fluctuations and that whole trees were exposed to the high temperature stress and not single leaves.

A great deal of variation in isoprene emission rates was observed. I was not convinced that the statistical treatments accurately reflected the variability. Isoprene emission is exceedingly difficult to predict, a point made by this lab that affects how these data need to be interpreted.

> **Reply:**
>
> As the reviewer states there is high variability in isoprene emission. This is due to changes in environmental drivers and tree-to-tree variability. However, besides this variability, differences between treatments were statistically significant as seen using linear-mixed effect models and uncertainty bounds for temperature response curves. Does the referee's comment further refer to the uncertainty bounds of the temperature and light fit, which might seem to be too low when compared to the variability of single measurement points? If this was the concern: as commonly done (see Seco et al., 2015) we used bin averaged data (each point weighted by the inverse of its standard deviation) to determine the fit for calculating the temperature and light response curves. The resulting confidence intervals of the fit (as shown in Figure 5) reflect that the fitted curves are statistically different above 25°C for heat trees and above 30°C for heat-drought trees (due to the higher variability in the heat-drought data). However, to clearly show that also the bin averaged

isoprene emissions $E_{iso}$ are significantly different between stress and control treatments for $T > 28°C$ we inserted Appendix B and Table B1 to a revised version of the manuscript.

While a lot of work has gone into this report, I have significant concerns including that leaf temperature is not known and measurement temperature during isoprene emission measurement was almost non-overlapping.

**Reply:**
We can understand the reviewer's concern about the missing leaf temperature measurements. We purposely decided against controlling leaf temperature and shortly explained our considerations in the last paragraph of Section 2.1.1. Additionally we included Appendix A and Table A1 which shows a comparison between air temperature and leaf temperature (which did not differ by more than 1°C).
A remark regarding the non-overlapping temperatures: The temperature curves of the control and heat-treated trees do not overlap because the control trees were not exposed to heat waves.

**Detailed Comments:**
Throughout the manuscript:

**Reply:**
All grammatical and style changes were implemented in a revised version of the manuscript.

Line 151: Does this mean that the air flow through the chamber was turned off when it wasn't being measured?

**Reply:**
When the chamber was not measured it was open and thus circulated with ambient air. This is explained in more detail in the first paragraph of section 2.1.1 in the revised version of the manuscript (line 166-172).

Line 275: Is this the same as Es?

**Reply:**
No. We explicitly introduced a parameter *EF* which was parametrized) for the temperature fit. We explained the issue in section 2.2, lines 293-299. However since we found that both *EF* and $E_S$ are basically identical we just used $E_S$ further on to avoid confusion.

Line 296: How was leaf water potential measured?

**Reply:**
We added the sentence '*Mid-day leaf water potential was measured by determining the pressure necessary to cause water to exclude from a freshly-cut leaf inserted in a Scholander pressure chamber (Model 1000, PMS Instrument Company, Albany, Oregon, USA).*' to Section 2.1.2 in the revised manuscript.

Line 300: This sounds as though the heat-drought trees were subject to a different experiment the previous year than were the other trees.

**Reply:**
To make clear that the trees of the heat and heat-drought treatment had been subjected to two heat waves in the previous year we added information to the experimental setup (line 113-118) and rewrote the corresponding sentence (line 359-364)

Line 317: It is hard to see this in 3c. The overwhelming impression is the variability.

**Reply:**
In this Figure, each data point reflects one measurement made with the automated leaf chamber, thus the variability includes diurnal variations and differences between the individual trees measured. The trees were exposed to ambient temperature changes and light fluctuations causing the pronounced variability in isoprene emissions the reviewer is referring to. We purposely decided to plot each single data point to highlight the difficulty of modeling isoprene emissions. However, alongside with the single data points we are also providing daily treatment averages, which clearly show the large increases in isoprene emissions. As the significance of the treatment effects have been confirmed in Table 2 of the revised manuscript we decided not to change Figure 3c as it just reflects the true variability in the data.

Line 355: Isoprene is very temperature dependent while photosynthesis is not but both a very light dependent. By restricting the analysis to less than 30oC the data would have been mostly in the light liming range for photosynthesis and so also for isoprene. Above 30°C light was probably mostly limiting and so the very different temperature responses of isoprene emission and photosynthesis would become dominant.

**Reply:**
We are not sure if we fully understood the reviewer comment. We assume that he/she refers to the differences in temperature responses of isoprene and photosynthesis which become more dominant for higher temperatures.
In the revised version of the manuscript we added the sentence '*The divergence between isoprene emissions and photosynthesis is most likely a consequence of the different temperature optima for isoprene emissions and photosynthesis. While above 30°C isoprene emissions are still increasing exponentially with temperature, photosynthesis is already decreasing (Ruehr et al. 2016) leading to the discrepancies between photosynthesis and isoprene emissions.*' to section 4.2 of the discussion.

Line 348: It isn't clear why this should be normalized to 500 when Es is normalized to 1000.

**Reply:**
Correct. In most studies Es is parameterized for light-saturation at 1000 µmol m$^{-2}$s$^{-1}$, however, the value used for standardization is an arbitrary value. In principle it does not matter to which light conditions Es is normalized as long as this value is above the light saturation for isoprene emissions. We added the sentence: '*Response curves of isoprene versus environmental conditions obtained in this study will thus be more comparable to lower canopy conditions where leaves are not constantly under light saturation. To reflect this, we normalized temperature responses of isoprene to a PAR of 500 µmol m$^{-2}$s$^{-1}$ (typically values of 1000 µmol m$^{-2}$s$^{-1}$ are used in the literature).*' to explain our consideration in Section 3.4 of the revised manuscript.

Line 353: I am not good at statistics, but I don't feel these numbers represent what I would intuitively take from Fig 5. The highest rate of the bin averaged control seems to exceed any control measurements. The table confirms that there was only one measurement above 32°C. The extreme variability, especially of the heat-drought treatment, make it difficult to draw specific

conclusions. I would only conclude that very high rates are possible in heat-drought but low rates are also possible, possibly reflecting dying leaves.

> **Reply:**
> The bin-averages 32–35°C and 35-40°C in the control treatment consist of one measurement point each. Since these points did pass the quality control, we did not exclude them from the fit, but used a lower weight which was calculated for each bin average using the inverse standard deviation (in case of n=1, SD was artificially set to 100; we added this to section 2.3 in the revised manuscript)
> In this case SE refers to the fit of EF, since the uncertainty of this parameter in the fit was relatively low (at least for the control and heat data) we got a relatively low standard error. Please consider that we did fit the curves to bin averaged data points and not to single measurement points (which are shown for reasons of transparency). We included Appendix B and Table B1 to a revised version of the manuscript which shows bin averages and corresponding standard errors to show that bin averaged isoprene emissions for $T > 28°C$ are significantly different (based on a t-test) between the control and the stress treatments as well.

Line 354: %a is a temperature response and not normalized data

> **Reply:**
> We apologize for this mistake and changed it in the revised version of the manuscript

Line 375: Both isoprene synthase and DMAPP availability affect this as recent papers have shown.

> **Reply:**
> Thank you for pointing us to this. We changed that accordingly in a revised version of the manuscript.

Line 386: First seen by LoretoSharkey TD, Loreto F (1993) Water stress, temperature, and light effects on the capacity for isoprene emission and photosynthesis of kudzu leaves. Oecologia95, 328-333.

> **Reply:**
> We added this reference in a revised version of the manuscript.

Line 397: I would also cite the work of Delwiche Delwiche CF, Sharkey TD (1993) Rapid appearance of 13C in biogenic isoprene when $^{13}CO_2$ is fed to intact leaves. Plant, Cell & Environment16, 587-591

> **Reply:**
> We added this reference in a revised version of the manuscript.

Line 404: Sharkey and Loreto saw 67% Sharkey TD, Loreto F (1993) Water stress, temperature, and light effects on the capacity for isoprene emission and photosynthesis of kudzu leaves. Oecologia 95, 328-333.

> **Reply:**
> We changed that accordingly in a revised version of the manuscript.

Line 436: I am not convinced of this

> **Reply:**

We have taken this information from the 95% confidence intervals of the fit derived for the control and the heat and heat-drought treatment (these do not overlap completely). In a revised version of the manuscript we added Fig 6 and Fig 7 which illustrate differences between measured data and the treatments fitted curves in in a better way.

Line 461: This is new and likely to well accepted but the current manuscript does not make a strong enough case for it.

**Reply:**
Thank you. As mentioned before, we modified some parts in the introduction, added Figures 6 and 7, Section 3.5, and Section 4.4 to highlight the novel point of this study.

**Anonymous Referee #3**

Major comments:
1) Materials and Methods, Experimental set up, line 96. I am concerned that the trees in the stress treatments had previously been exposed to two experimental heat waves and were showing a difference in basal area. Previous work has shown that VOC emissions differ based on exposure to previous environmental conditions (e.g. Sharkey et al, 1999 and citing references). Could the authors provide some reassurance that after pruning and over wintering the development and growth rates were then equivalent and could be fairly compared to one another? If they were not equivalent as suggested in the results section 3.1, were the data normalised?

**Reply:**
The data were collected as part of a full three-years experiment which sought to evaluate the response to prolonged and repeated stress. During the first year it was unfortunately not possible to collect VOC data, but information from the first year of the experiment showed that black locust leaves recovered its photosynthesis 3 weeks after the last heat wave ended and that basal growth rates were close to control trees (Ruehr et al. 2016). Trees were pruned due to height constraints in greenhouse facility and overwintered outside. We improved this explanation in Section 2.1. lines 113-118.
Before leaf-out in spring, the trees were returned inside the greenhouse and equipped with sensors. Branch chambers were installed in June. Statistical analysis showed no differences in leaf gas exchange (net photosynthesis and isoprene emission) before the heat-waves were imposed in the second year of the experiment (see Table 2 of the revised manuscript). We explicitly stated that before stress net photosynthesis and isoprene emissions did not differ significantly among treatments (line 373-377 and line 382-385). Therefore we do not think it would be necessary to normalize the data and we are confident that leaf level emissions did not carry a substantial signal as a consequence of the stress during the first year of the experiment.

2) Could the authors give an explanation as to why the trees were not randomly selected for the work included in the current study? This would have given a mixture of previously stressed and unstressed trees in each treatment group and removed any concern that the prior treatment of these trees was affecting the current results.

**Reply:**

This is an important point, and reflects the study design over the entire duration. The purpose was to evaluate how the trees will response to re-occurring heat waves over subsequent years – which made it necessary to maintain trees within one treatment (now explained in Section 2.1. lines 113-118).

Studies on heat waves occurring over more than one growing season are scarce and to our knowledge have not been done yet with woody species. Although we found a slightly reduced basal area of previously heat and heat-drought stressed trees in the second year of the experiment, we detected no change in leaf-level emissions of newly grown leaves of stressed trees compared to the control prior to the second year heat waves' (see LME results in Table 2 of the manuscript).

3) I would also like to see a clear description of the growth conditions and number of trees used per treatment and per measurement. Could the authors give a full description of the growth conditions of the trees (temperature, light, CO2, RH) in the description of the experimental setup? Did the greenhouse have supplemental lighting, where was average PAR recorded, what was the day length? How many replicates were used per measurement? At the moment it is not clear to me how many replicates were used for what.

**Reply:**

The reviewer is correct in that we did not provide all this information in the Methods section, but instead referred the reader to a publication that describes the experimental set-up in great detail. In order to facilitate reading of the manuscript, we added more detail on the methods into a revised manuscript (mainly in Section 2.1 and 2.1.1)

Minor comments
1) Abstract line 12 – mentions assessing the impact of stress on BVOC emissions but only isoprene is presented in the manuscript. Either remove the reference to general BVOC or include other emitted compounds.

**Reply:**

We changed the wording accordingly

2) Intro, line 38 – include ref to more recent Wyche et al, ACP 2014 which gives positive and negative effects of isoprene emission on secondary aerosol formation.

**Reply:**

Thank you for pointing us to this reference. We included it in a revised version of the manuscript

3) Into, line 65 and line 71 – include ref to more recent Ryan et al, New Phyt 2014 and remove older references unless they are seminal /original work.

**Reply:**

We included the more recent literature and removed some of the older literature where it was appropriate.

4) Mat and Methods, Paragraph starting line 155 – description is not clear. Is the automatic switching of the measurements or the air flow? If air flow does this mean the chambers were clamped on the plants with no air flow for a period of time?

> **Reply:**
> Thank you for pointing us to this shortcoming of our methods description. This is now better explained in section 2.1.1, lines 166-172.

5) Section VOC Line 200 – the PTR-MS only counts set masses and cannot give compound identification. Could the authors include information on any mass identification that was performed (e.g. GC-MS) to confirm that it was only isoprene at m/z 69

> **Reply:**
> It is true that the PTR-MS only counts nominal masses. Since black locust is known to be a relatively strong isoprene emitter we are confident that in our case, as well as in other studies (see Vanzo et al., 2015) the signal on m/z 69 is due to isoprene. This is now better explained in Section 2.1.3, lines 247-254.

6) Line 231 – 500 PAR seems quite low for trees in the summer. Top of canopy PAR in northern Europe during the summer is more likely to be between 1000 and 2000 PAR. Could the authors give a reason for choosing 500 PAR.

> **Reply:**
> Correct. In most studies $E$s is parameterized for light-saturation at 1000 $\mu$mol m$^{-2}$s$^{-1}$, however, the value used for standardization is an arbitrary value. In principle it does not matter to which light conditions $E$s is normalized as long as this value is above the light saturation for isoprene emissions. We added an explaining sentence to revised version of the manuscript (line 414-417).

7) Mat & Methods Line 267 - Formatting error

> **Reply:**
> This was corrected in the revised version of the manuscript.

8) Results 3.1 line 295. Could the authors include a description of how midday leaf water potential was measured?

> **Reply:**
> We added this information to a revised version of the manuscript (line 230-232).

9) Results 3.1 line 299 – typo "relative" should be "relatively"

> **Reply:**
> Thank you for catching this. The typo was corrected in a revised version of the manuscript.

10) Results 3.2 line 307 – I don't understand why " (PAR > 50 umol m-2 s-1)" is included in this sentence, when the sentence is referencing stomatal conductance – please clarify.

> **Reply:**
> We clarified that we explicitly calculated daytime averages.

11) Line 316 Daytime (PAR > 50 umol m-2 s-1) – I am assuming this means the authors collated any data collected when PAR readings were over this value to be "daytime" values. If this is correct please include a clarification at first use to make it easier for the reader to understand.

> **Reply:**
> We clarified that in the revised version of the manuscript.

12) Line 322 – It may be over-stretching the results to include "marginally significant (p value around 0.1)" results as significant differences. This is not common practice but is perhaps personal preference.

> **Reply:**
> We wanted to indicate that the p-value suggests that these values tend to be higher compared to the control even if the change is not significant based on the $p<0.05$ criterion and changed the wording accordingly.

13) Results 3.3, line 338 "significantly different to control trees" and "no significant differences. . ." please give p values.

> **Reply:**
> Agreed. We added the corresponding choice for a significant change (p-value < 0.05) to the data analysis section.

14) Discussion Line 380 – references you should include more recent ref e.g. Ryan et al New Phyt 2014 who used genetically modified tobacco specifically to study the impact of drought on isoprene emission and protection.

> **Reply:**
> Thank you. We made sure to include the more recent literature in the revised version of the manuscript.

15) Line 385 "A quick recovery of isoprene emissions after periods of drought stress seems to emerge as a 385 common feature that has also been observed in previous studies (Brilli et al., 2013; Pegoraro et al., 2004; Velikova and Loreto, 2005)" and line 288 "The observed faster recovery of isoprene emissions than photosynthesis may be a common pattern following stress release (Brilli et al., 2013; Pegoraro et al., 2004)." This appears to be a repeated point – please remove one of the sentences.

**Reply:**
We reworded that part of the discussion (line 467-470) to avoid repetitions.

16) Line 390 – "this is the first study that considers dynamics of isoprene emissions during and following combined heat–drought stress. . ." Unfortunately this claim is untrue
– please remove and see Vanzo et al, 2015 and references therein.

**Reply:**
We apologize for this mistake and reworded the sentence accordingly and discussed our study with respect to the results therein (line 471-482)

17) Paragraph beginning line 415 – including reference to Ryan et al, 2014, New Phytologist, who studied isoprene emitting and non-emitting plant responses to drought,
would be appropriate here. Most likely with the Vickers et al, 2009 reference.

**Reply:**
Done as suggested.

18) Table2–could the authors explain why there is such a variation in group sizes (n values from 0–49)?

**Reply:**
We explained that in Section 2.3, lines 331-337.

**References:**

Boeck, H. J. De, Dreesen, F. E., Janssens, I. A. and Nijs, I.: Climatic characteristics of heat waves and their simulation in plant experiments, Glob. Chang. Biol., 16, 1992–2000, doi:10.1111/j.1365-2486.2009.02049.x, 2010.

[revised manuscript text omitted]